# Ground ice, organic carbon, and soluble cations in tundra permafrost soils and sediments near a Laurentide ice divide in the Slave Geological Province, N.W.T., Canada

Rupesh Subedi[1], Steven V. Kokelj[1, 2], and Stephan Gruber[1]

[1]Department of Geography and Environmental Studies, Carleton University, Ottawa, ON, K1S 5B6, Canada
[2]Northwest Territories Geological Survey, Yellowknife, NT, X1A 2L9, Canada
**Correspondence:** Stephan Gruber (stephan.gruber@carleton.ca)

**Abstract.** The central Slave Geological Province is situated 450–650 km from the presumed spreading centre of the Keewatin Dome of the Laurentide Ice Sheet, and it differs from the western Canadian Arctic, where recent thaw-induced landscape changes in Laurentide ice-marginal environments are already abundant. Although much of the terrain in the central Slave Geological Province is mapped as predominantly bedrock and ice-poor, glacial deposits of varying thickness occupy significant portions of the landscape in some areas, creating a mosaic of permafrost conditions. Limited evidence of ice-rich ground, a key determinant of thaw-induced landscape change, exists. Carbon and soluble cation content in permafrost are largely unknown in the area. Twenty-four boreholes with depths up to ten metres were drilled in tundra north of Lac de Gras to address these regional gaps in knowledge and to better inform projections and generalizations at coarser scale. Excess-ice contents of 20–60 %, likely remnant Laurentide basal ice, are found in upland till suggesting that thaw subsidence of metres to more than ten metres is possible if permafrost were to thaw completely. Beneath organic terrain and in fluvially-reworked sediment, aggradational ice is found. The variability in abundance of ground ice poses long-term challenges for engineering, and it makes the area susceptible to thaw-induced landscape change and mobilization of sediment, solutes and carbon several metres deep. The nature and spatial patterns of landscape changes, however, are expected to differ from ice-marginal landscapes of western Arctic Canada, for example, based on greater spatial and stratigraphic heterogeneity. Mean organic-carbon densities in the top 3 m of soil profiles near Lac de Gras are about half of those reported in circumpolar statistics, deeper deposits have densities ranging from 1.3–10.1 $kg\,C\,m^{-3}$, representing a significant additional carbon pool. The concentration of total soluble cations in mineral soils is lower than at previously studied locations in the western Canadian Arctic. This study can inform permafrost investigations in other parts of the Slave Geological Province and its data can support scenario simulations of future trajectories of permafrost thaw. Preserved Laurentide basal ice can support new ways of studying processes and phenomena at the base of an ice sheet.

# 1 Introduction

A unique drilling program sampling permafrost in the tundra north of Lac de Gras resulted in 24 boreholes with depths up to ten metres. It sampled the active layer and permafrost layer of soils and sediments and allowed investigating their contents of ground ice, organic carbon and soluble cations. These three interrelated topics (e.g., Littlefair et al., 2017; Lacelle et al., 2019) are relevant for understanding the nature of permafrost and for anticipating consequences of its thaw, which are expected to become increasingly persistent and widespread due to anthropogenic global climate change.

The Lac de Gras region (Fig. 1), as part of the Slave Geological Province, is of interest because its geomorphic setting and Quaternary history differ from more intensively studied areas in the previously glaciated western Canadian Arctic and in unglaciated terrain in Yukon and Alaska (Dredge et al., 1999; Karunaratne, 2011). Its Holocene periglacial evolution spans only about 9,000 years, it is situated relatively close to an ice divide of the Laurentide Ice Sheet. Even though ice divides shift over time (Margold et al., 2018; Boulton and Clark, 1990b), predominant zones of erosion and deposition by the Laurentide Ice Sheet, and likely previous ice sheets, are apparent at the continental scale and have been linked with continental patterns of ice flow and basal thermal regime (Sugden, 1977, 1978; Boulton and Clark, 1990a). The area beneath the Keewatin Dome spreading centre (Fig. 1C, zone 1), the predominant location of its ice divides, was characterized by low subglacial erosion rates and often has thick till. Areas near the margin of the ice sheet are frequently characterized by high deposition rates and many of these environments with thick till are relatively well accessible and studied in the western Canadian Arctic (Fig. 1C, zone 3). Between both (Fig. 1, zone 2) there is evidence for an area of increasing glacial erosion and basal conditions transitioning from melting to refreezing to fully frozen. The Slave Geological Province largely falls into this intermediate zone that is characterized by predominantly thin glacial sediments, and mineral soils are often coarse and locally sourced from igneous and metamorphic rocks. The conditions in this zone likely result in high spatial and stratigraphic heterogeneity in the landscape, creating the need for detailed study of permafrost conditions and careful scaling approaches for coarse-scale models.

Several mines in the area as well as the planned Slave Geological Province Corridor (all-season road, power transmission, communication) add applied relevance in the long term. This billion-dollar infrastructure project will connect Yellowknife with mines and future mineral resources in the study area and may eventually connect Canada's highway system to a deep-water port on the Arctic Ocean in Nunavut. The study presented here has been enabled by the Slave Province Surficial Materials and Permafrost Study, a large partnership of industry, government and academia.

The ice content of permafrost strongly determines the consequences of thaw such as subsidence or thermokarst development. It thereby also controls potential damage to infrastructure as well as the amount and timing of carbon fluxes into the atmosphere (Turetsky et al., 2019) and nutrient release into terrestrial and aquatic ecosystems (Lantz et al., 2009; Kokelj et al., 2013). The surroundings of Lac de Gras are shown as continuous permafrost with low (0–10 %) visible ice content in the upper 10–20 m and sparse ice wedges in the Permafrost Map of Canada (Heginbottom et al., 1995) and are designated as having thin overburden cover (<5–10 m) and exposed bedrock in the Circum-Arctic Map of Permafrost and Ground-Ice Conditions (Brown et al., 1997). For both, it is the lowest class of ground-ice content in continuous permafrost. The new Ground Ice Maps for Canada (O'Neill et al., 2019) show the study area (50 km × 50 km) to contain no or negligible wedge ice, negligible to low

segregated ice and no relict ice, which includes buried glacier ice. By contrast, the hummocky tills that cover about 20% of the study area have been hypothesized to contain large ice bodies, possibly of glacigenic origin (Dredge et al., 1999) as proposed also in other areas (e.g., Dyke and Savelle, 2000). Improving our understanding of the vertical distribution, spatial heterogeneity, and characteristics of ground-ice are a key prerequisite for better simulating and anticipating the consequences of permafrost thaw.

Large stocks of organic carbon that can be decomposed and transferred to the atmosphere upon thaw (Schuur et al., 2008) are held in permafrost (Hugelius et al., 2014). The integration of organic carbon into the near-surface permafrost is related to either periods of deeper thaw, which can redistribute carbon within the soil profile, or to a rising permafrost table due to colluviation or alluviation, peat accumulation, or climate cooling. These processes affect both carbon and geochemical profiles (Lacelle et al., 2019). To support the generation of future climate scenarios, the quantification and characterization of permafrost organic

carbon storage is important and little information on soil organic carbon exists within a large area surrounding Lac de Gras, especially at depths exceeding one metre (Hugelius et al., 2014; Tarnocai et al., 2009).

Nutrients, organic materials and contaminants (natural and anthropogenic) can be released from permafrost during thaw (Dyke, 2001; Leibman and Streletskaya, 1997; Mackay, 1995), translating geomorphic disturbance, forest or tundra fire, or atmospheric warming into impacts on the chemistry of soils and surface water, and provoking noticeable ecological and down-

stream effects (e.g., Frey and McClelland, 2009; Kokelj and Burn, 2005; Kokelj et al., 2009; Littlefair et al., 2017; Malone et al., 2013; Tank et al., 2016). Studies from northwestern Canada report permafrost, the transient layer and the active layer to have distinct physical and geochemical characteristics (Kokelj et al., 2002; Kokelj and Burn, 2003, 2005; Lacelle et al., 2014) and sometimes distinguish relict/paleo-active layers (Burn, 1997; Lacelle et al., 2019). These vertical patterns are attributed to (a) past thawing causing loss of ground ice, leaching of solutes from thawed soils and redistribution of organic carbon by

cryoturbation (Kokelj and Lewkowicz, 1999; Kokelj et al., 2002; Leibman and Streletskaya, 1997; Pewe and Sellmann, 1973), and (b) thermally-induced moisture migration during soil freezing redistributing water and soluble ions (Cary and Mayland, 1972; Qiu et al., 1988) contributing to solute enrichment in near-surface permafrost (Kokelj and Burn, 2005). These study areas, however, are different from the Slave Geological Province. For example, the alluvial materials derived from sedimentary and carbonate rock of the Taiga plain together with regular flooding produce solute rich active layer and permafrost deposits in

the Mackenzie Delta. As another example, the sediments that comprise Herschel Island are silty-clay tills that include coastal and marine deposits excavated by the Laurentide Ice Sheet (Burn, 2017). In contrast to previous findings in these areas, we hypothesize that the tills in the Lac de Gras region are solute poor because they are locally sourced from granitic bedrock (Hu et al., 2003) and had limited potential for chemical weathering at depth. This is in line with sediments of similar origins, but contrasting depositional and permafrost history near Yellowknife reported to have low soil solute concentrations (Gaanderse

et al., 2018) with variable trends in vertical profiles suggestive of active layer leaching and signs of evaporative concentration depending on site history (Paul et al., 2020).

This study aims to improve the understanding and quantitative characterization of permafrost and active layer materials in tundra environments near Lac de Gras and contribute to better understanding permafrost environments in the intermediate zone between the margins and the Keewatin Dome of the Laurentide Ice Sheet in the Slave Geological Province more broadly.

The objectives are to *(i)* quantify the amounts and vertical patterns of excess ice, organic carbon and total soluble cations, *(ii)* explore factors contributing to the variation in physical and chemical characteristics between terrain types, and *(iii)* compare excess-ice content, organic-carbon density and total soluble cation concentration with other permafrost environments or with compilations such as overview maps and databases. We interpret multiple boreholes grouped by terrain type and, with the data available, distinguish unfrozen (active layer) and frozen (predominantly permafrost) samples but do not additionally separate

transient or relict active layers.

## 2   Study region

The study region (110.3° W, 64.7° N) is north of Lac de Gras, approximately 200 km south of the Arctic Circle and about 310 km northeast of Yellowknife. The regional climate is continental, with summers cool and short, and winters cold and extremely long (Hu et al., 2003). Ekati, a diamond mine in the study region, has a mean annual and summer air temperature of -8.9

100 °C and 14.0 °C, respectively, and an annual precipitation sum of 275 mm during 1988–2008 (Environment Canada, 2019). Deglaciation occurred before 8,500 BP and between 6,000 and 3,000 BP, forest tundra extended to approximately the study area and then retreated again (Dyke, 2005; Dredge et al., 1999).

The region is in the zone of continuous permafrost (Figure 1) and mapped as having low (0–10 %) visible ice content in the upper 10–20 m (Heginbottom et al., 1995; Brown et al., 1997). One map indicates sparse ice wedges and the other thin

overburden (<5–10 m) with exposed bedrock. A recent circumpolar compilation of permafrost carbon data (Hugelius et al., 2014) estimated soil organic-carbon storage (SOCs) to be 5–15 (0–1 m) and 15–30 $kg\ C\ m^{-2}$ (0–3 m). Recent work in the area has produced a wealth of permafrost stratigraphic (Gruber et al., 2018a) and thermal (Gruber et al., 2018b) data that enabled not only this contribution but also several simulation studies (Cao et al., 2019a; Melton et al., 2019; Cao et al., 2019b) predicting permafrost temperature driven by global atmospheric models.

For spatial context, we consider a 50 km × 50 km study area, and additionally, its surroundings as characterized by the 1:125,000 National Topographic System (NTS) of Canada maps on surficial geology 'Lac de Gras' (76-D, Geological Survey of Canada, 2014b) and 'Aylmer Lake' (76-C, Geological Survey of Canada, 2014a). These map areas are located 450–650 km from the presumed mean spreading centre of the Keewatin Dome and about 100–300 km from the transition of thick to thin glacial sediments that is apparent on coarse-scale maps. It generally is a source area for sediments, unlike ice marginal

locations. The spatial abundance of surface materials and previously predicted relict ice content is summarized in Table 1 (see also Fig. S1 in the Supplement).

The study area is characterized by low relief where irregular bedrock knobs and cuestas form hills up to 50 m high (Dredge et al., 1999). The northern part is dominated by till deposits, whereas the southern half consists more prominently of bedrock with patches of till (Hu et al., 2003). Numerous eskers and outwash complexes, mostly composed of sand and gravel, are found

in the area (Dredge et al., 1994). Till deposits are differentiated by their estimated thickness into till veneer (<2 m thick), till blanket (2–10 m thick), and hummocky till (5–30 m thick). These deposits typically have a silty sand to sand matrix with low

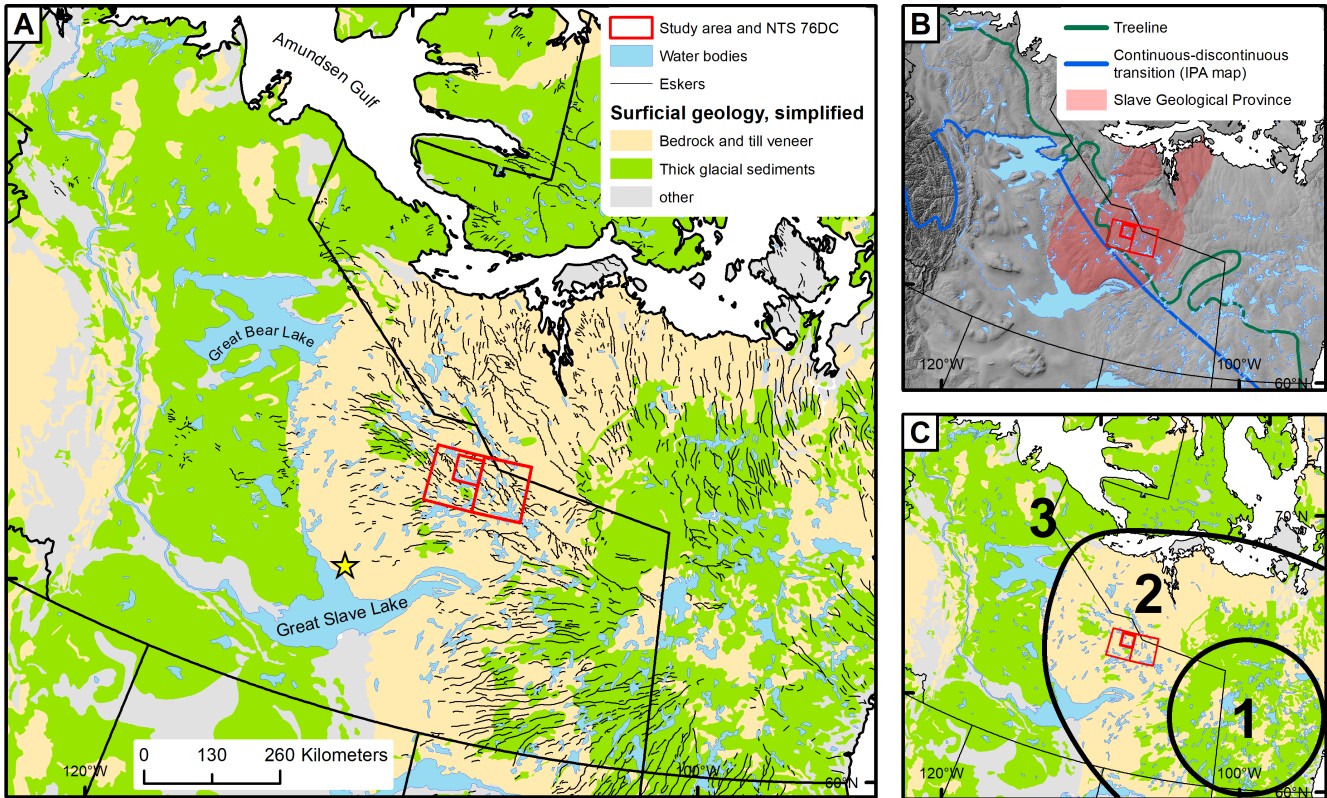

**Figure 1.** (A) Physiographic context of the 50 km × 50 km study area near Lac de Gras and the surrounding map sheets 76-D and 76-C. The simplified surficial geology legend distinguishes zones of thin and of thick till. Eskers indicate approximate late-glacial ice flow. (B) Location of the study area with respect to treeline, the mapped transition between continuous and discontinuous permafrost, and the Slave Geological Province. (C) Three zones of differing thickness of glacial sediment are apparent, zone (1) is assumed to correspond to the location of the Keewatin Dome spreading centre, zone (2) is an assumed area across which glacial erosion increases and basal conditions transition from melting to refreezing to fully frozen, and zone (3) is near the margin of the ice sheet, frequently characterized by high deposition rates and thick till. Data: Surficial Geology Map of Canada (Geological Survey of Canada, 2014c), Geological Map of Canada (Wheeler et al., 1996) and CanVec Hydro Features; Northern Canada Geodatabase (1.0); Circum-Arctic map of permafrost and ground-ice conditions (Brown et al., 1997). Yellowknife is the closest city and indicated with a yellow star.

percentages of clay and 5–40 % gravel (Wilkinson et al., 2001). Overburden thickness is considerable in hummocky till and till blanket of the study area (Haiblen et al., 2018) and the two surrounding map sheets (Kerr and Knight, 2007).

Soil parent materials consist of till, glacio-fluvial sediments, or peat. Upland till surfaces are characterized by mud boils, earth hummocks and organic material, observed to depths of up to 80 cm, that has been redistributed within the active layer by cryoturbation (Dredge et al., 1994). The tills derived from granitic and gneissic terrain have a silty or sandy matrix, whereas those derived from metasedimentary rocks contain a higher silt-clay content (Dredge et al., 1999). Low-lying areas are mostly comprised of colluvium or alluvium rich in organics and wet areas often have polygonal peatlands (Karunaratne, 2011).

**Table 1.** Spatial abundance of lakes, surficial geology and estimated relict ground ice for the 50 km × 50 km study area and surroundings (Fig. 1). Surficial geology is based on the 1:125,000 map sheets 76-D and 76-C, percentages are relative to exposed land area, whereas values in square brackets are relative to total surface area including lakes. The abundance of relict ground ice is from O'Neill et al. (2019), who use a model based on data products at the scale of 1:5,000,000.

| | study area | NTS 76-D | NTS 76-C |
|---|---|---|---|
| area (km$^2$) | 2,500 | 10,706 | 10,709 |
| Lakes (%) | [40] | [28] | [29] |
| Bedrock (%) | 12 [8] | 13 [10] | 8 [5] |
| Glaciofluvial (%) | 3 [2] | 2 [1] | 1 [1] |
| Till veneer (%) | 29 [17] | 34 [24] | 19 [13] |
| Till blanket (%) | 32 [19] | 40 [28] | 49 [35] |
| Hummocky till (%) | 19 [11] | 5 [4] | 17 [12] |
| Organic (%) | 5 [3] | 6 [5] | 6 [5] |
| Relict ice: none (%) | 100 | 96.1 | 99.9 |
| Relict ice: negligible (%) | 0 | 0 | 0 |
| Relict ice: low (%) | 0 | 3.8 | 0 |
| Relict ice: medium (%) | 0 | 0.1 | 0.1 |
| Relict ice: high (%) | 0 | 0 | 0 |

The area is in continuous shrub tundra (Wiken et al., 1996) and common shrubs include northern Labrador tea *(Rhododendron tomentosum)* and dwarf birch *(Betula glandulosa)*, while bog cranberry *(Vaccinium vitis-idaea)* and dwarf bog rosemary *(Andromeda polifolia)* often comprise the understory (Karunaratne, 2011). Well-drained upland areas are typically covered with a thin layer of lichens and mosses (Hu et al., 2003) (Figure 2A). Grasses and sedges with a ground cover of moss comprise the vegetation cover in valleys (Figure 2B) and some poorly-drained low-lying areas have thick peat associated with ice-wedge polygons and sedge meadows (Hu et al., 2003; Karunaratne, 2011) (Figure 2C). Frequently, low-lying areas have tall shrubs along small streams and at the rise of steeper slopes. Esker tops have little vegetation and are often comprised of exposed soil (Figure 2D).

## 3  Field observation and sampling

In summer 2015, soil cores with a nominal diameter of 5 cm, but often irregular due to partial melting and reaming, were obtained using a diamond drill (Kryotek Compact Diamond Sampler), sectioned into 20 cm intervals and logged (soil texture, colour, ice content and visible organic matter) in the field while still frozen (Subedi, 2016; Gruber et al., 2018a). Esker locations

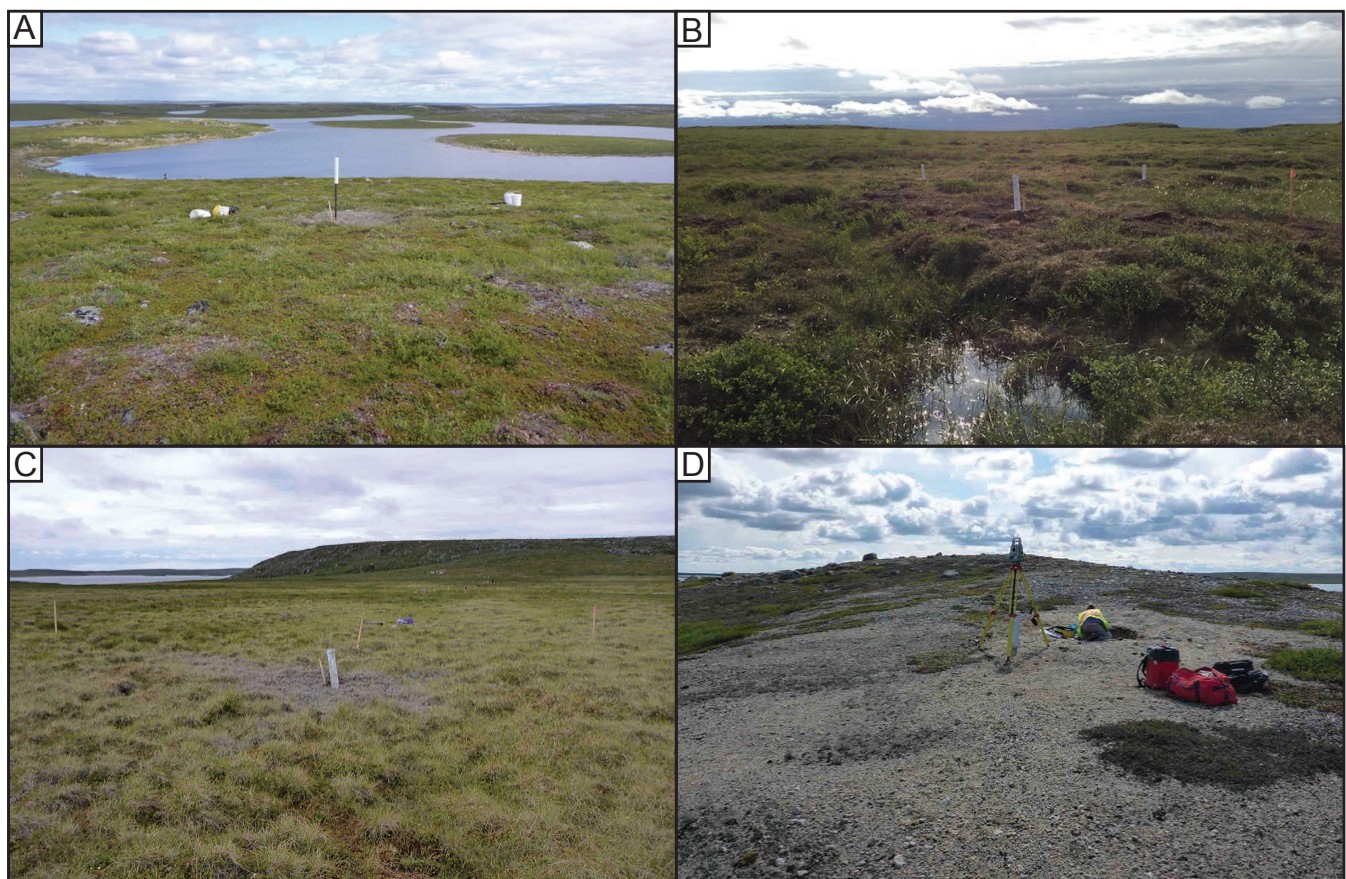

**Figure 2.** Examples of the four terrain types used: (A) upland tills, (B) organic deposits, (C) the Valley, and (D) eskers.

were augered. Two soil pits were excavated within approximately 10 m of each borehole to describe and sample typical near-surface soil conditions. Where possible, samples were taken at depths of 10 cm, 20 cm, and 30 cm; exact sample volumes are not known. The depth of thaw at the time of sampling was estimated by probing, although this was often unsuccessful in coarse mineral soil. Drill core and pit samples were double-bagged for thawed shipment to the laboratory in Yellowknife.

Borehole locations were originally planned for investigating (a) vertical and spatial patterns of solute and organic-carbon content in soils, and (b) terrain effects on ground temperature based on thermistor chains installed after drilling. For site selection, we used the surface classes of the 1:125,000 surficial geology map 76-D, topographic position, aerial imagery revealing surface cover such as vegetation or boulders, and a Landsat derived index outlining the location of late-lying snow drifts. Locations were planned in clusters to simplify the logistics of moving equipment with a helicopter. The reverse-circulation winter
drilling campaign during March and April 2015 (Normandeau et al., 2016) inspired the selection of site NGO-DD15-1014, where a thick sequence of ice rich ground was encountered.

During fieldwork, blocky surfaces could not be sampled as moving clasts jammed the drill rods. Sections with high gravel content or boulders resulted in slow progress and were often terminated at relatively shallow depth due to time constraints. Furthermore, these sections often resulted in low recovery, because the heating of the drill when cutting through hard rock would melt the frozen core and fines would be washed out. The complete drill logs and photographs revealed a cluster of boreholes with well-graded fine sands and pronounced ice lenses as well as till with high excess ice content beneath upland locations. These clusters, however, do not correspond well with the surficial mapping units used. Correspondingly, four terrain types (Figure 2) comprised of Upland Tills, fluvially reworked till (the Valley), Organic terrain and Eskers are used as a grouping for describing and interpreting the drill cores and soil pits at 24 locations (Table A1, see also Fig. S1 in the Supplement). The uneven depth and sparse sampling within boreholes led us to report results from multiple boreholes in combined plots).

Upland Tills: Ten boreholes were sampled to depths of 2.5–9.5 m in smoothly rounded hills comprised of thick till and in till veneer over bedrock. The dominant cryostructure was wavy and suspended. The dominant plant species were dwarf shrubs, Labrador tea and grasses. Thaw depths were about 2 m on hill tops and nearly a meter at the bottom of hills.

The Valley: Eight boreholes were drilled to depths of 1–6 m in a gently sloped valley that contrasts with other terrain types because its silts and sands are well sorted and likely derived from fluvial reworking of local tills. Boreholes located on the more elevated sides of the Valley typically had coarser sediments, whereas those near its axis had mostly fine sediments with high silt contents and organics with ice-wedge polygons. The dominant cryostructure was lenticular. Few water logged sites contained tall shrubs with water channels. Sites were sparsely to moderately covered with plant species such as dwarf birch, Labrador tea and grasses. Thaw depths were 35–40 cm.

Organic terrain: Two boreholes to depths of about 4.5 m were drilled on the centres of ice-wedge polygons in peatlands with hummocky surface topography. The dominant plant species were dwarf birch and Labrador tea, with plenty of low-lying grasses. The depth of thaw was 35 cm and the permafrost table 50–70 cm deep.

Eskers: Four boreholes were drilled to depths of 1.5–12 m at hilltop locations with sparse vegetation or exposed soil.

## 4 Methods

All samples were thawed and processed at ambient temperature. Samples were homogenized, poured into beakers, weighed, and allowed to settle for 12 h (cf. Kokelj and Burn, 2003). Volumes of sediment $V_s$ and supernatant water $V_w$ were recorded to estimate volumetric excess ice content (%) of the permafrost samples as

$$V_{ei} = \frac{1.09\,V_w}{V_s + 1.09\,V_w} \times 100, \tag{1}$$

where 1.09 approximates the density of water divided by that of ice. The volumetric percentages of coarse fragments (>5 mm), sand (0.074–5 mm) and fines (<0.074 mm) in the sediment was estimated visually to the nearest 5 %. The volumetric percentage of coarse fragments ($V_c$) relative to the total sample volume was obtained by multiplying the estimated coarse percentage with $1 - V_{ei}/100$. The length of solid rock cored per borehole was estimated from the core photographs.

Supernatant water was extracted directly from samples where sufficient volume was available and to all other samples, a known amount of deionized water was added (1:1 extraction ratio; Janzen, 1993). These samples were mixed thoroughly and

then allowed to settle for 12 h. Water was collected with a syringe and filtered through 0.45 $\mu$m cellulose filter paper. The remainder of the sample was dried for 24 h at 105 °C to determine the gravimetric water content (%), expressed on a dry basis ($GWC_d$) and on a wet basis ($GWC_w$) (cf. Phillips et al., 2015).

The concentration (mg/l) of the soluble cations Ca++, Mg++, Na+ and K+ was determined by atomic adsorption spectrophotometer at the Taiga lab in Yellowknife. Measured soluble ion concentrations $C^m$ (mg/l) were converted to an expression $E$ using milli-equivalents per unit mass of soil (meq/ 100g of dry soil)

$$E = \frac{C^m}{M^e} \times M_w^{100g}, \tag{2}$$

where $M^e$ is the equivalent mass of ions (g) and $M_w^{100g}$ is the mass of water per 100 g of dry soil as present in the sample at the time of water extraction. Presentation of soluble cation concentrations per unit mass of dry soil facilitates comparison between samples of varying moisture contents. We sum the resulting four soluble cation concentrations to obtain total soluble cation concentration.

Organic-matter content $LOI$ (%) is expressed on a gravimetric dry basis and was determined using the sequential loss-on-ignition method (Sheldrick, 1984) at Carleton University. A small amount (2–3 g) of the homogenized and oven dried sample (< 0.5 mm soil fraction) was placed in a crucible and heated to 550 °C for 6 h to determine the organic-matter content as

$$LOI = \frac{M_S^{105} - M_S^{550}}{M_S^{105}} \times 100, \tag{3}$$

where $M_S^{105}$ is the mass of sediment after oven drying at 105 °C, and $M_S^{550}$ is the mass of sediment after ignition at 550 °C. Because homogenization involved crushing the oven-dried sample with mortar and pestle, any larger organic fractions like roots and plant remains are therefore part of the sample analyzed for LOI. To avoid combustion problems, reduced amounts (0.5–1 g) were processed when samples consisted of plant residue with very little visible mineral soil. When no mineral soil component was visible after coarse components were removed, samples were not processed and an LOI of 80% was estimated. This occurred only in the top 0.3 m and almost exclusively in samples from soil pits. The gravimetric percentage ($P_{0.5}$) of the <0.5 mm soil fraction has been lost from the original analysis. Later, this was determined again based on dry sieving for 183 of 357 samples.

Data quality was assessed in a second analysis on the samples using the same procedures and tools as during the original processing. Based on measured blanks, the accuracy is about 0.03 % LOI, the median accuracy based on doubles is 0.04 % LOI with the highest difference being 0.30 % LOI.

Soil organic-carbon storage (SOCs, $kg\,C\,m^{-2}$) was computed for comparison with soil carbon inventories (e.g., Hugelius et al., 2014). For this, soil organic-carbon concentration (SOCc, % mass) was computed as

$$SOCc = \frac{LOI}{2.13}, \tag{4}$$

following Dean (1974) and dry bulk density (DBD, $kg\,m^{-3}$), which is known to correlate with SOCc (e.g., Alexander, 1989; Bockheim et al., 2003), was approximated as

$$DBD = 71 + 1322 \times e^{(-0.071 \times SOCc)}, \tag{5}$$

following Hossain et al. (2015), who conducted their study in geologic settings similar to the project area. This resulted in an estimated DBD for the fine-grained soil, i.e. excluding the volumes $V_{ei}$ and $V_c$. To account for this, soil organic-carbon density (SOCd, $kg\ C\ m^{-3}$) was derived as

$$SOCd = \frac{SOCc}{100} \times \frac{P_{0.5}}{100} \times DBD \times \left(1 - \frac{V_{ei} + V_c}{100}\right), \tag{6}$$

and finally applied as average values over depth intervals within each terrain type to obtain SOCs. Aggregated SOCd and SOCs were reduced by the average proportion of solid rock cored though in Upland Tills and in Organic terrain below 2 m. For the samples without measured values, $P_{0.5}$ was estimated by beta regression (Cribari-Neto and Zeileis, 2010) with $LOI$, $GWC_w$ as well as the visually-estimated proportions of sand and fines as independent variables. Predictors are significant at the 0.1 %-level and residuals have a standard deviation of 11 %.

Estimating $P_{0.5}$ for 176 of 357 samples statistically, parameterizing $DBD$ and estimating $V_c$ visually introduce uncertainty in the resulting values for $SOCd$ and $SOCs$. The potential magnitude of this effect on average values is estimated by computing a low-carbon scenario and a high-carbon scenario by varying $V_c$ and $P_{0.5}$ by $\pm 10$ percentage points each and $DBD$ by $\pm 50\ kg\ m^{-3}$. These deviations are subjective choices and correspond to a bias of twice the recorded precision for $V_c$ and approximately the standard deviation of model residuals for $P_{0.5}$. The deviation for $DBD$ is chosen to be considerable while within the variation observed by Hossain et al. (2015, Fig 6). The averages of the resulting scenario values are 37 % lower and 38 % higher than the best estimate for SOCd that is reported and interpreted in the following sections.

Detailed grain-size distribution was measured on selected samples using a Beckman Coulter LS 13320 laser-diffraction particle-size analyzer. Samples were first oven-dried at 105 °C and then crushed and homogenized with a mortar and pestle. Samples were then passed through a 2 mm sieve to remove the coarse fraction that was then weighed. Organic matter was removed from the fines using hydrogen peroxide. The samples were then mixed with Calgon to prevent flocculation and passed through the particle-size analyzer. Results were classified according to the USDA textural classification system (2 mm > sand > 53 $\mu$m > silt > 2 $\mu$m > clay).

## 5 Results

### 5.1 Soil texture

Approximately 7 % of upland tills (3.4 of 49.4 m of borehole depth, Table A1), consist of rock clasts larger than the core barrel. Eskers were augered rather than cored and large clasts would have terminated the borehole and not be recovered. The majority of the Valley cores were free of large clasts. Most soils consisted of poorly to very poorly sorted silt and sand. The relative proportion of silt was high in samples from mineral soils beneath organic terrain and in valley bottom sites with average values exceeding 40 %. Clay content was low and always below 20 %.

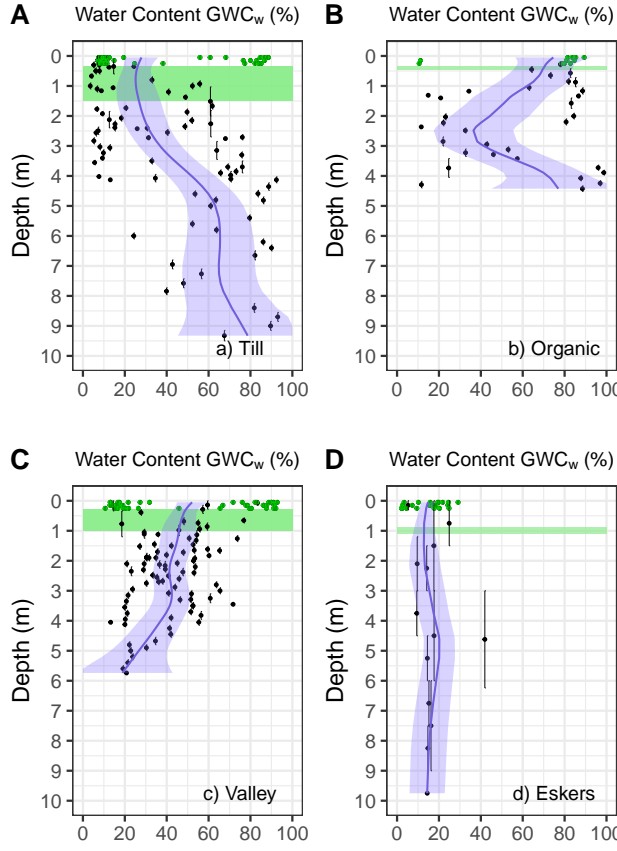

**Figure 3.** Gravimetric water content from four terrain types near Lac de Gras, N.W.T. Black points and their vertical lines represent borehole samples and their depth intervals, the number and individual characteristics of boreholes are in Table A1 and in the Supplement. Green points show pit samples. Blue lines represent averaged values, taking into account sample depth intervals, with shaded blue areas indicating the standard error at 95% confidence. Green shaded areas indicate the range of thaw depths for the boreholes at the time of sampling.

## 5.2 Ground ice

Field logged visible-ice content is available for 113 core sections. The average, weighted by the length of core sections, is 24 %. Cryostructure is discussed in Section 6. Details about individual boreholes are in the Supplement and all core photographs are available in Gruber et al. (2018a). Laboratory analyses show that water and excess-ice contents increase progressively

with depth in Upland Till. Zones of high moisture content (Figures 3A and 4A) were often associated with ice lenses, several centimeters thick (e.g., Figures B1 and B3). Excess-ice content greater than 50 % in Upland Till became increasingly common below 4 m depth; 5 boreholes terminated in ice-rich material and 5 in rock. In organic terrain, high moisture content (>80 %) but low excess-ice content in permafrost reflect saturated organic soils with low bulk density (Figure 3B). The sharp decline in water content below 2 m depth coincides with a decline in organic matter contents (Figure 4B). A notable increase in moisture

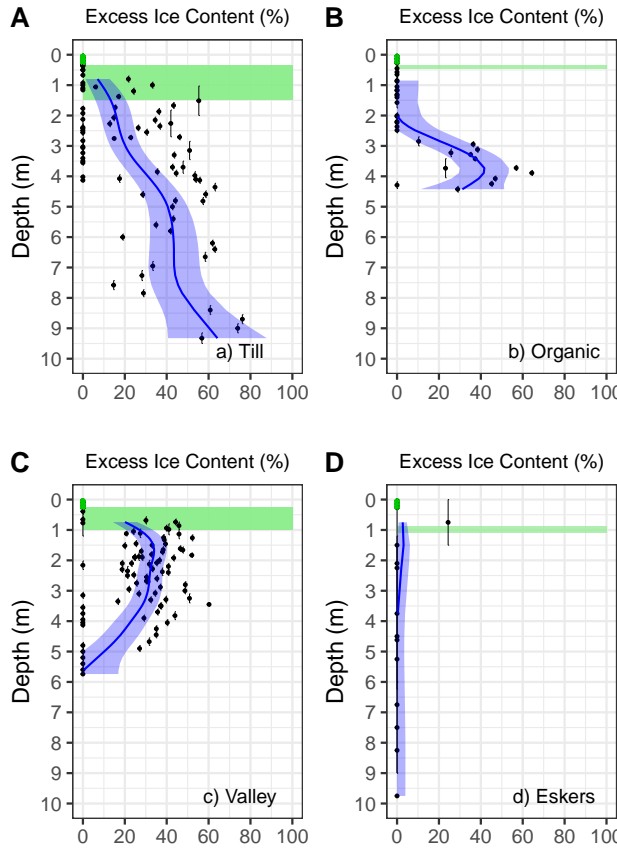

**Figure 4.** Excess-ice content from four terrain types near Lac de Gras, N.W.T. Black points and their vertical lines represent borehole samples and their depth intervals, the number and individual characteristics of boreholes are in Table A1 and in the Supplement. Green points show pit samples. Blue lines represent averaged values, taking into account sample depth intervals, with shaded blue areas indicating the standard error at 95% confidence. Green shaded areas indicate the range of thaw depths for the boreholes at the time of sampling.

and excess ice content from 2.5–4 m depth occurred in underlying mineral soils. Profiles from the Valley showed high moisture content near the surface, where organics were present, and deeper down (Figure 3C) due to 20–50 % excess ice in mineral soil (Figure 4C). Frozen and ice poor till has been recovered near the bottom of boreholes in Organic terrain and in the Valley (Figure B4). In eskers, water content was mostly below 20 % and pore ice was the dominant ground-ice type (Figure 3D).

### 5.3 Organic carbon

Organic-carbon density in the active layer was typically greater than at depth in permafrost (Figure 5). Statistics of soil organic-carbon density and storage are given for consistent depth intervals and the four terrain types in Table 2.

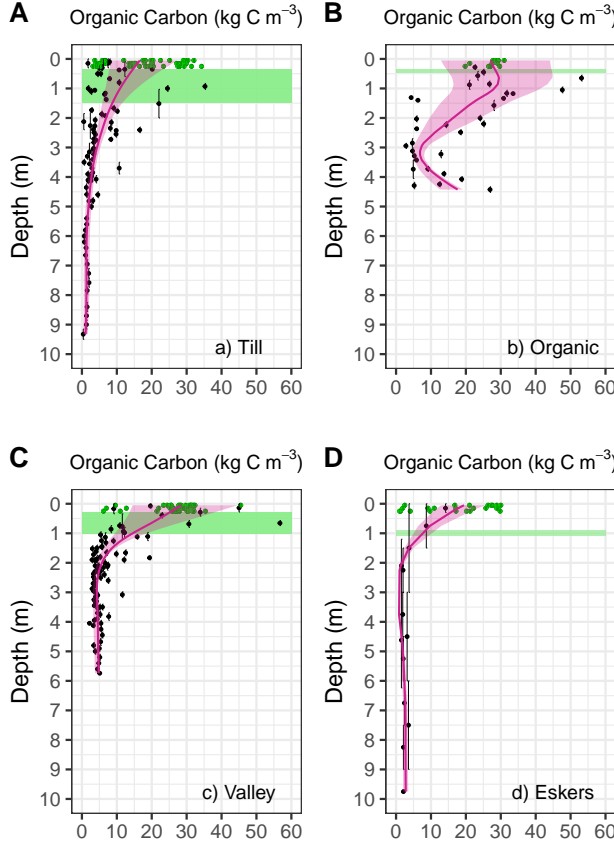

**Figure 5.** Soil organic-carbon density ($kg\,C\,m^{-3}$) from four terrain types near Lac de Gras, N.W.T. Black points and their vertical lines represent borehole samples and their depth intervals, the number and individual characteristics of boreholes are in Table A1 and in the Supplement. Green points show pit samples with approximated values for the organic terrain. Magenta lines represent averaged values, taking into account sample depth intervals, with shaded magenta indicating the high/low-carbon scenarios used to estimate the uncertainty inherent in estimating and parameterizing some of the values used in calculations. Green shaded areas indicate the range of thaw depths for the boreholes at the time of sampling.

### 5.4 Total soluble cations

The concentrations of soluble cations (meq /100g dry soil) in organic-rich, shallow soils were mostly higher and more variable than those in mineral permafrost soils at depth (Figure 6). In organic materials, the concentration of soluble cations near the top of permafrost was relatively high (Figure 5B and 6B). In Upland Till, soluble cation concentrations, as with ice content, increased gradually with depth (Figure 4A and 6A). Differences between active layer and permafrost, as well as between organic and mineral soils are apparent from their median concentration of soluble cations; organic samples are distinguished using a threshold of 30 % LOI (cf. CSSC, 1998) and permafrost considered when logged as frozen. In organic samples the

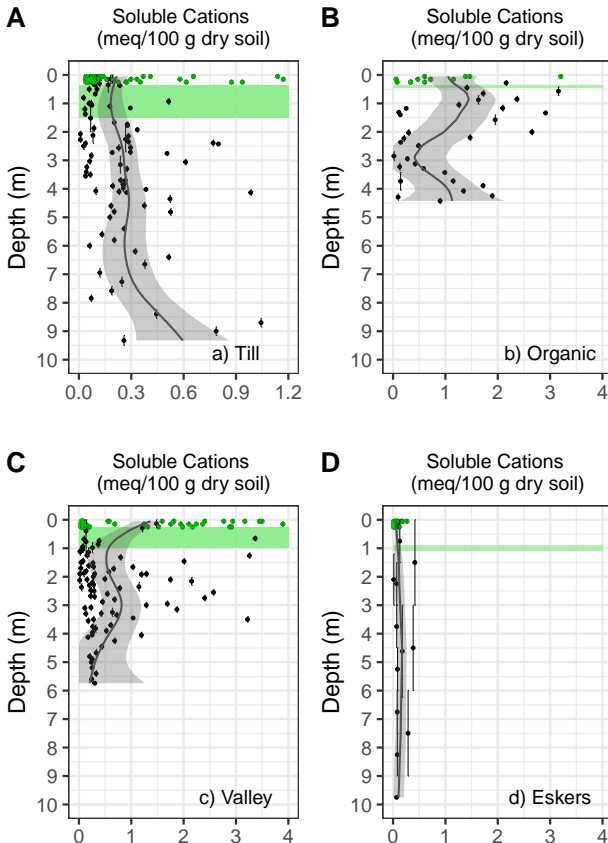

**Figure 6.** Total soluble cation concentration (meq/100 g dry soil) from four terrain types near Lac de Gras, N.W.T. Black points and their vertical lines represent borehole samples and their depth intervals, the number and individual characteristics of boreholes are in Table A1 and in the Supplement. Green points show pit samples. Smooth black lines represent averaged values, taking into account sample depth intervals, shaded grey areas indicate the standard error at 95% confidence. Green shaded areas indicate the range of thaw depths for the boreholes at the time of sampling. Note differences in horizontal scales.

contrast (permafrost to active layer) was 2.02 to 0.34 meq/100 g dry soil and in mineral samples 0.26 to 0.09 meq/100 g dry
soil. The four group medians are all significantly (p < 0.01) different from each other based on Kruskal-Wallis tests. Although the dry bulk density of organic soil is lower than that of mineral soil, these patterns persist even when expressed relative to wet soil mass (p<0.01) or the volume of water contained in the thawed sample (p<0.05).

**Table 2.** Mean observed soil organic-carbon density (SOCd, $kg\,C\,m^{-3}$) per depth interval and soil organic-carbon storage (SOCs, $kg\,C\,m^{-2}$) for the four terrain types investigated. SOCs is based on average SOCd accumulated from the surface down to the specified maximum depth. Ranges in parentheses indicate minimum and maximum values rounded to the nearest integer, the number of samples is indicated in square brackets. Values for Upland Tills and at depths below 2 m in organics account for the abundance of rock clasts larger than the core barrel. The top 0.3 m are subject to high uncertainty arising from the uniform estimation of 80 % LOI for organic-only samples.

| Depth | Organics SOCd | SOCs | The Valley SOCd | SOCs | Upland Till SOCd | SOCs | Eskers SOCd | SOCs |
|---|---|---|---|---|---|---|---|---|
| 0–0.3 m | 26.5 (19–31) [13] | 8 | 26.5 (6–45) [52] | 8 | 14.8 (2–31) [59] | 4 | 17.5 (1–30) [25] | 5 |
| 0.3–1 m | 29.9 (21–53) [5] | 29 | 20.6 (8–57) [8] | 22 | 10.9 (4–33) [10] | 12 | 8.7 (9–9) [1] | 11 |
| 1–2 m | 25.6 (4–48) [9] | 54 | 7.6 (3–19) [23] | 30 | 7.8 (2–23) [13] | 20 | 3.7 (4–4) [1] | 15 |
| 2–3 m | 11.8 (3–23) [8] | 66 | 4.8 (3–8) [22] | 35 | 5.4 (1–15) [16] | 25 | 1.8 (2–2) [2] | 17 |
| 3–5 m | 10.1 (4–25) [11] | 87 | 4.8 (2–12) [25] | 44 | 2.7 (1–10) [23] | 31 | 2.2 (2–3) [3] | 21 |
| 5–10 m | | | | | 1.3 (0–3) [16] | 37 | 2.4 (2–4) [5] | 34 |

# 6 Interpretation and discussion

## 6.1 Ground ice

The results presented are subject to a number of biases compared to a perfectly randomized sampling within each terrain type and complete recovery of samples during drilling. Drilling induced errors in the recovery of ground ice where excessive heating of the drill barrel caused partial or complete thaw of the core (Figure 7). Depending on the degree of thaw and core composition, this resulted in intervals erroneously shown with reduced or no excess ice content (Figure 4C). The results are, therefore, likely to be conservative (low biased) estimates of excess ice and gravimetric water content at the locations sampled.

The difficulty of drilling though large clasts, on the other hand, may have caused bias towards sampling locations with higher contents of excess ice and of fines. When drilling organics within polygon networks, the drill rig was placed on polygon centres. As a consequence, wedge ice, which is known to be present in the area based on the surface expression of polygon networks, is systematically avoided in sampling and, therefore, largely excluded from the present quantitative data and interpretation.

While this study was not designed to elucidate the origin of ground ice, a number of observations merit discussion. In
organic terrain, the excess ice recovered resembles pool ice (clear with small bubbles and embedded peat filaments, Figure B2B) and wedge ice (foliated with bubbles and some sediment, Figure B2C), as previously reported for polygonal ground in organics (Mackay, 2000; Morse and Burn, 2013). In the Valley and in Organic terrain, the increase in water and excess-ice content with depth (Figures 3 and 4) is likely due to ice segregation in frost susceptible and relatively well graded mineral soil (Figures B1 and B2 panel D) with frequent reticulate cryostructure. Both the Valley site, comprised of materials that are
likely to be fluvially-reworked tills, and Organic terrain represent aggradational environments in low lying areas where water tends to accumulate and the terrain surface gained material either through the accumulation of peat or fluvial deposition. Near the surface, permafrost aggraded upwards into Holocene deposits in these settings, and as such, does not contain relict or

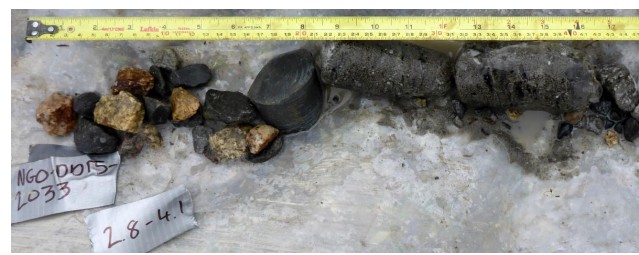

**Figure 7.** Heating of the drill barrel, likely due to cutting through the rock near the middle of this section, caused complete melt of ground ice in the left part of the core shown. Here, gravel was recovered but fines were likely washed out with melt water and drilling fluids. The right side of the core is partially thawed and ice rich. This recovered section of less than 0.5 m represents 1.3 m in borehole NGO-DD15-2033.

preserved ice within the depth of thaw prior to and during aggradation. Similarly, permafrost has aggraded in the sediment of eskers, where less fine material together with well-drained convex topography explains the absence of aggradational ice. Eskers in the study area occasionally contain relict ice, interpreted as partially derived from glacial meltwater and deposited together with the esker sediments (Dredge et al., 1999; Hu et al., 2003) and show geomorphic evidence of melt-out (Prowse, 2017).

Cores from upland hummocky till, analyzed to 9 m depth (Figure B3), were often associated with high amounts of excess ice. Ice occurred in wavy layers and sometimes as clear ice several centimetres thick (similar to Fig. 3.8c in Gruber and Haeberli, 2009). In contrast to the Valley site and Organic terrain, no reticulate structure and no apparent separation of consolidated fines and clear ice were visible in Upland Till cores. Sediment was coarser and more poorly sorted in Upland Till than at the Valley site or beneath Organics. Particles appeared to be suspended in the ice matrix, giving it a visual appearance of much lower ice content than it actually has, as reported previously for basal glacier ice facies (Knight, 1997; Murton et al., 2005). Soluble cation contents are lower in mineral soil active layers and near-surface permafrost than at depth, consistent with near-surface leaching and preservation of underlying materials in the frozen state since deglaciation. Finally, the convex topography of Upland Till sites makes them well drained and differentiates them from the Valley and from Organic sites. These differences in cryostructure, cation content and sedimentological properties point to ice of differing origin and deposit type. The presence of hummocky topography, thermokarst lakes and thaw-slump like features (Fig. C1), suggest melt-out of ice-rich till, and the involuted or hummocky nature of some hilltop surfaces resemble other permafrost preserved glaciated landscapes in northwestern Canada, which are known to host relict Pleistocene ground ice (Dyke and Savelle, 2000; Rampton, 1988; St-Onge and McMartin, 1999). By contrast, Rampton (2000) invoked subglacial hydrology rather than ground-ice melt in interpreting features such as the one shown in Fig. C1, calling them (inverted) plunge pools that were caused by scouring where pressurized turbulent water was forced to change direction. Both, ground ice characteristics and geomorphic features suggests that a large proportion of the excess ice in this hummocky till is Laurentide basal ice preserved beneath ablation till.

The excess ice contents encountered in mineral soil were often 20–60%. As a first-order estimate, this implies that complete thaw of permafrost can cause about 0.2–0.6 m of subsidence for each vertical meter. The boreholes in upland hummocky till show an increasing trend of excess ice content with depth, based on our limited sampling to 9 m, alone. Half the boreholes

terminate in ice rich ground and the other half in rock. Not having boreholes terminate in ice poor till indicates that thicker sequences of ice rich material than what has been recovered can be expected. Furthermore, an earlier winter drilling campaign (Normandeau et al., 2016) produced six boreholes in upland tills. NGO-RD15-150, co-located with NGO-DD15-1014, logged 13 m of 'ice' with bedrock at 16.7 m. The other boreholes (NGO-RD15-148, 155, 160) only returned minor ice content with depths between 5.5 and 7.9 m. While this provides additional context, the absence of logged ice needs to be interpreted with care give the combined difficulties of logging reverse circulation recovery and winter conditions, and because logging ice content has not been a priority of that campaign. Surface lowering of several meters, with potential of up to more than ten metres, could thus be expected from areas of thick upland till if this permafrost was to thaw completely. This includes the potential for thermokarst processes to mobilize sediments, solutes and organic carbon at depth more quickly than expected in strictly conductive one-dimensional thaw. A number of geomorphic features reminiscent of retrogressive thaw slumps (Fig. C1) and the presence of kettle lakes (Prowse, 2017) in the area both exhibit local relief that suggest meltout of massive ice several metres in thickness.

### 6.1.1 Glaciological context

In the interior of the Laurentide Ice Sheet, no supraglacial sources of debris existed. If the ice found in Upland Tills is indeed Laurentide basal ice, then its mineral content must derive from basal entrainment. Debris rich basal ice can result from a variety of processes that occur at glacier beds with net freezing or net melting (Alley et al., 1997; Cuffey et al., 2000). Some of the processes of freezing involve the migration of liquid water akin to permafrost aggradation and ice lensing studied in periglacial environments (Christoffersen and Tulaczyk, 2003).

Using ice samples from the Greenland Ice Sheet, Herron et al. (1979) showed debris-laden ice in the lowermost 15 m beneath a divide and Souchez et al. (1995) showed vertical mechanical mixing of clean ice and basal ice beneath the summit of the ice sheet. Furthermore, they pointed to the similarity of the anomalously high gas content ($CO_2$ and $CH_4$) to that in permafrost soils, invoking the mobilization of soils predating the ice sheet. The original basal ice prior to upward mixing has partially formed at the ground surface and is hypothesized to be a remnant of the growing stage or the original build up of the ice sheet (Souchez et al., 1994). Assuming vertical mixing near the bed of the ice sheet, the lowermost meters of ice may thus be composed to varying degrees of ice derived from precipitation and metamorphism at the surface of the ice sheet and of materials formed from the freezing of liquid water in debris (cf. Gow et al., 1979). The hypothesized basal ice in the project area, however, is richer in sediment, contains coarser clasts, and is thicker than that reported from ice cores beneath the summit of Greenland (Gow et al., 1997), especially when accounting for previously thawed material that overlies the remnant ice found today.

Most studies of basal ice from ice sheets originate from either ice coring at modern ice divides, modern margins of Arctic ice caps that have preserved basal ice-sheet ice (LeB. Hooke, 1976), or from studies near the margins of former ice sheets, where the presence of buried basal ice in permafrost (Murton et al., 2004) is well established. The present study may provide the first evidence of basal ice in the zone a few hundred kilometers from ice divides (as conceptualized in Figure 1C, Zone 2) where rates of erosion increase and the thermal regime at the base varies (Sugden, 1977, 1978; Boulton, 1996). These conditions

are described by Hooke et al. (2013) who reconcile glaciological theory with observations from mineral prospecting. They predict the formation of thick dispersal plumes in the transition zone from basal melting to basal refreezing to a fully frozen bed, that is likely to have occurred in the study area. Glacial sediment plumes are thus likely composed of directly eroded bedrock incorporated into the basal ice by, and together with, refreezing basal melt water. Additionally, regolith and organics predating the ice sheet may be incorporated. The model of Hooke et al. (2013) is useful in explaining why we find a spatial patchwork of preserved excess ice. First, the thermal conditions at the bed of the ice sheet likely had a patchwork character in this transitions zone, making the distribution of sediment plumes uneven. Second, basal ice can only be preserved where it is overlain by sediment thicker than the maximum depth of thaw during the Holocene. As spatial patterns of plume thickness, mineral content, and vertical structure are expected to vary, so will the likelihood of preserving ice until today.

### 6.1.2 Estimated spatial abundance of relict ice and implications for mapping and modelling at coarse scale

Although our results suggest relict ice preserved in areas of thick upland till, they do not easily support quantitative conclusions on the spatial extent over which this occurs. Nevertheless, we can constrain the range of plausible extent based on a few assumptions. Let us express the areal proportion underlain by some amount of relict ice as $P_{ice} = P_{up} \times P_{pre}$, with $P_{up}$ being the proportion of upland in a particular class of till and $P_{pre}$ being the proportion of area within the upland portion where preserved ice is found. Based on visual interpretation of maps and imagery, we assume $P_{up}$ to be 30–70 % in hummocky till and 5–30 % in till blanket. We can attempt to estimate $P_{pre}$ based on what we interpret as preserved ice at three drilling locations (Figure S1, NGO-DD15-1014 and around NGO-DD15-2004 and NGO-DD15-2033) and not finding it during winter drilling at three other locations (NGO-RC15-148, 156, 160). Winter drilling with reverse circulation may produce false negatives, whereas one could argue that NGO-DD15-2004 and NGO-DD15-2033 are close and should not be counted separately. With these biases, the low number of locations sampled, and the implications of varying sediment-plume characteristics in mind, let us assume $P_{pre}$ to be 25–75 %. Correspondingly, $P_{ice}$ is 7–53 % for hummocky till and 1–23 % for till blanket, considering till veneer and bedrock are negligible by comparison. This leads to estimated areal proportions of the land surface underlain by some relict ice of 2–17 % for the study area, 1–12 % for NTS sheet 76-D, and 2–20 % for NTS sheet 76-C (Table 1).

Even though the areal proportion underlain by relict ice is uncertain, the contrast with the recent map of O'Neill et al. (2019) is striking (Table 1). It highlights the importance of spatial heterogeneity and of representing it at coarse scale, where the dominant surface type alone may not be a good predictor of actual relict ice content, or ice content in general. This scale mismatch is a common challenge for permafrost models (Gruber, 2012) and more generally for models of non-linear processes (Giorgi and Avissar, 1997). The rules proposed by O'Neill et al. (2019) would likely predict relict ice in the project area similar to observations if used with the surficial geology at 1:125,000 scale. By contrast, the study area has been generalized into the classes of till veneer and bedrock, exclusively, in the surficial geology map 1:5,000,000 that was used as input for the nation wide map of O'Neill et al. (2019). In earlier permafrost maps and models, uncertainty and variability at the sub-grid scale have been propagated into model results with high/low cases on assumed sub-grid processes and with the presentation of additional instructions for field interpretation (e.g., Boeckli et al., 2012; Gruber, 2012). Analogously, introducing a scaling relationship between the surficial geology at 1:125,000 and 1:5,000,000 would allow to apply the rules of O'Neill et al. (2019)

at the national scale while improving the representation of heterogeneity. For example, a fractional sub-grid proportion of hummocky till in till veneer at 1:5,000,000 could be assumed. Extending this to other classes and introducing best, high and low estimates would make the scaling issue a part of the modeling process and propagate its effects into the final results.

## 6.2 Organic carbon

Terrain variation in organic-carbon density and in organic matter content in fine material occur in association with surficial material and topographic setting. Organic terrain is characterized by peat deposits up to 2.5 m thick and associated with low lying, poorly drained portions of the landscape. In other terrain types, organic materials may have become vertically redistributed in the top few metres of soil profiles by cryoturbation (Dredge et al., 1994; Kokelj et al., 2007; Haiblen et al., 2018) and by burial during permafrost aggradation due to colluviation/alluviation (e.g., Lacelle et al., 2019). The low organic-

carbon density at depth likely indicates the absence of sediment reworking and permafrost preservation in upland tills during the Holocene.

Mean organic-carbon density in the top 3 m of soil profiles near Lac de Gras is about half that of the circumpolar mean values reported in a recent compilation for similar soils (Table D1). For mineral soils, this is similar to the mean of about 12 ($kg\,C\,m^{-3}$) in the northern Canadian Arctic and for organics, it is similar to the mean of about 30 ($kg\,C\,m^{-3}$) in the southern

Canadian Arctic reported for the top 1 m by Hossain et al. (2015, Fig. 5D). The low organic-carbon density in the study area, especially at depth, is interpreted to derive from the short duration of Holocene carbon accumulation following at least partial evacuation of older soil carbon by the Keewatin sector of the Laurentide Ice Sheet. While deep carbon pools are important (Koven et al., 2015), corresponding data, as reported here, is rare (Hugelius et al., 2014; Tarnocai et al., 2009). Values reported here on organic-carbon density in the top 0.3 m are subject to high uncertainty arising from the uniform estimation of 80%

LOI for organic-only samples. This is because peat usually has higher (Treat et al., 2016) and non-peat organic material lower (Hossain et al., 2015) LOI.

## 6.3 Total soluble cations

In mineral soil, the lower concentration of total soluble cations in the active layer (median of 0.05 meq/100 g dry soil) compared with permafrost (median of 0.25 meq/100 g dry soil) is interpreted to be caused by leaching of ions from unfrozen soil, and

410 is similar to observations in other regions (Table D2). This large contrast between mineral soil active layers and permafrost is likely robust even though variable extraction ratios were used. Fitting a model to predict total soluble cation concentration in mineral soil from $GWCw$ and the extraction ratio shows $GWCw$ as a highly significant predictor, while the extraction ratio is not significant (p>0.05). Additionally, the redistribution of ions along thermal gradients during freezing may have caused solute enrichment during the development of segregated ice (Figure B1 and B2) in aggrading permafrost (cf. Kokelj and Burn,

2005) in mineral soils of the Valley and beneath peat in organic terrain. There, zones of increased cation concentrations at depth corresponded with ice-rich intervals in permafrost , especially at sites in till and in organic terrain (Figure 4 and 6). Where high amounts of organic matter are present, also the concentration of total soluble cations is high. As a consequence, mineral-soil

permafrost has lower concentrations of soluble cations than organic active-layer soils but higher concentrations than mineral active-layer soils.

The absolute concentrations of soluble cations obtained in the study area near Lac de Gras are low compared to previous studies from northwestern Canada that report higher concentrations in active layer and permafrost across diverse terrain types (Table D2). In the Mackenzie Delta, alluvial materials derived from sedimentary and carbonate rock of the Taiga plain and regular flooding produce solute rich active layer and permafrost deposits. A range of forest-terrain types contained more soluble cations, often several times higher, in the active layer and permafrost than the mineral soils in this study. Also in
comparison with undisturbed terrain on Herschel Island, the absolute concentrations of soluble cations in our study are low. Sediments on Herschel Island are silty-clay tills that include coastal and marine deposits excavated by the Laurentide Ice Sheet (Burn, 2017). These materials can be saline below the thaw unconformity indicating permafrost preservation of soluble materials below the maximum depth of early Holocene thaw (Kokelj et al., 2002) or their concentration in colluviated materials (Lacelle et al., 2019). The low concentrations in our study area are associated with the contrasting nature and origins of surficial
materials. Tills in our study region are generally coarser grained than many glacial deposits studied in the western Arctic, are regionally sourced from mostly granitic rocks and in many upland locations have been exposed to only minor postglacial landscape modification (Haiblen et al., 2018; Rampton and Sharpe, 2014). Although analytical methods are different, studies near Yellowknife (Gaanderse et al., 2018; Paul et al., 2020) also suggest low soluble cation concentrations in materials of similar origins but with contrasting depositional environments, terrain history and ecological conditions. Water from glaciolacustrine
delta sediments west of Contwoyto Lake, about 120 km north of the study area, has been reported with an average concentration of 6.2 meq/l for 8 samples from 3–12.4 m (Wolfe, 1998), similar to the average value of 7.5 meq/l resulting from 10 Esker samples of our study in the same depth range.

    The values reported must be interpreted in light of the ambiguity involved in the choice of method for extracting water from samples and in normalizing analytical results for comparison. Non-uniform extraction ratios can add uncertainty due to
increased uptake of solutes from soil where water content is high (Toner et al., 2013). This effect is difficult to prevent with multiple samples having $GWCw$ larger than 90% and standardization by extraction from a saturated paste as used in soil science is impractical with large clasts and high ice content. As such, the values obtained provide an imperfect comparison of the total cations that can be dissolved from the sediment within each sample. For comparison of samples, analytical results from permafrost studies are often normalized to dry soil mass (common in soil science) or water volume (common in studies
of water chemistry or glaciology), either that contained in the sample or that used during extraction of solutes. Because no uniformly accepted protocol exists, comparison between studies is often challenged by differences in their methods.

    Sections with relatively high solute concentrations exist in several boreholes. We hypothesize that those are, at least partially, caused by fresh rock flour produced when the diamond drill bit cuts through clasts, e.g., in NGO-DD15-2006 near 2.4 m depth. As such, the summarized concentrations we report may have a high bias.

## 7 Conclusions

The research area near Lac de Gras is characterized by a mosaic of terrain types with a high degree of fine-scale spatial variability in subsurface conditions. Permafrost there contains much more ground ice, slightly less organic carbon and fewer soluble cations compared with national and global compilation products or published research from sites in the western Canadian Arctic. This study provides quantitative data in a region with few previous studies and it supports six specific conclusions:

1. Excess-ice contents of 20–60 % are common, especially in samples from upland till and till-derived sediments, and the average field logged visible-ice content is 24 %. This new regional insight improves upon coarse-scale compilations that map the area north of Lac de Gras as ice poor (O'Neill et al., 2019; Brown et al., 1997; Heginbottom et al., 1995). Specifically, it points to the importance of scaling issues when applying models with coarse-scale input data.

2. Thick occurrences of excess ice found in upland tills are likely remnant Laurentide basal ice, and aggradational ice is found beneath organic terrain and in fluvially-reworked till.

3. Thaw-induced terrain subsidence on the order of metres to more than ten metres is possible in ice-rich till. Even though this study did not investigate the spatial abundance of ice rich till, it can be estimated as 2–17 % for the study area. Organic terrain hosts wedge ice and is typically underlain by ice-rich mineral deposits. Future thermokarst processes may therefore result in significant landscape change and fast mobilization of sediment, solutes and carbon to a depth of several metres. Geomorphic evidence of past ground-ice melt, including thaw-induced mass wasting exists.

4. Peatlands were found to be up to 2.5 m thick and in till and till-derived deposits, cryoturbation and colluviation/alluviation have redistributed modest amounts of organic carbon locally to depths of 2–4 m. Mean organic-carbon density in the top 3 m of soil profiles near Lac de Gras is about half that reported in recent circumpolar statistics (Hugelius et al., 2014).

5. The concentration of total soluble cations, expressed as meq/100 g dry soil, in active layer and permafrost mineral soils is markedly, often by one order of magnitude, lower in the Lac de Gras area than at other previously studied locations in the western Canadian Arctic. Mineral-soil active layers have a lower concentration of total soluble cations than permafrost. Total soluble cation concentrations are higher where soils are rich in organic matter.

6. Abundant relict ground ice and glacigenic sediments exist at locations in the interior of the Laurentide Ice Sheet and are poised for climate-driven thaw and landscape change, similar to permafrost-preserved ice-marginal glaciated landscapes where major geomorphic transformations are already observed (e.g., Kokelj et al., 2017; Rudy et al., 2017). The characteristics of thaw-driven landscape change, however, are expected to differ from observations in ice-marginal positions due to differences in (a) topography and climate affecting location and timing, (b) geotechnical properties affecting stability and mobility of sediments, and (c) geochemistry affecting solute and carbon release to surface water, ecosystems and the atmosphere.

These findings highlight the importance of geological and glaciological legacy in determining the characteristics of permafrost and the potential responses of permafrost systems to disturbance and climate change. The existence of preserved

Laurentide basal ice offers a unique chance to better study processes and phenomena at the base of an ice sheet. This opportunity will gradually diminish as the ice can be expected to progressively melt in the future. This future melt will partially reveal subsurface conditions through the nature and magnitude of change. Continued research on permafrost and landscape response to warming at locations in the interior of the Laurentide Ice Sheet will help to understand and predict changes specific to these landscapes characterized by a mosaic of contrasting permafrost conditions, and how they affect ecology, climate, land use and infrastructure.

*Code and data availability.*  Drill logs, visible ice content and core photos from the 2015 campaign are published (Gruber et al., 2018a). Data and code for reproducing the main figures, constructed using GG-plot in R, are available at https://doi.org/10.5281/zenodo.3628070

*Author contributions.*  SG and RS wrote the manuscript together with SVK. RS conducted the initial study, performed or oversaw the laboratory analyses, and produced the scripts for plotting Figures 3–6. SG produced Figure 1, the Supplement, the sections on ground ice, glaciology, soil organic-carbon density as well as framing and conclusions.

*Competing interests.*  No competing interests are present.

*Acknowledgements.*  This research was part of the Slave Province Surficial Materials and Permafrost Study (SPSMPS) supported by the Canadian Northern Economic Development Agency, Dominion Diamond Mines and the Northwest Territories Geological Survey. Additional support was obtained from the Natural Sciences and Engineering Research Council of Canada and ArcticNet. We thank Barrett Elliott and Dr. Kumari Karunaratne for their great support in this project and Dr. Chris Burn for his advice. We thank Julia Riddick and Rosaille Davreux for their help in soil sampling; Nick Brown, Luca Heim and Christian Peart for field assistance; Jerry Demorcy and Dr. Elyn Humphreys for their help in laboratory analysis; Cameron Samson for helping with LiDAR data; and the Taiga lab in Yellowknife for their assistance with the laboratory analysis of the samples. We acknowledge the help of Shintaro Hagiwara and the Carleton Centre for Quantitative Analysis and Decision Support with data smoothing in the profile figures, and Ariane Castagner and Nick Brown for sorting-out the samples for reprocessing. Interactive comments by two anonymous referees and by Dr. S. Wolfe have helped to improve this manuscript. This is NTGS Contribution #0130.

## Appendix A:  Sample locations

Table A1 details on borehole locations and sampling dates.

**Table A1.** Borehole locations and drilling dates, detailed plots are contained in the Supplement (Fig. S1–3). Full ID codes have the prefix 'NGO-DD15-'. Map units correspond to surficial geology at the scale of 1:125,000 (Geological Survey of Canada, 2014b). The depth and material at the end of hole (EOH) are indicated. The length of rock drilled though is estimated to the nearest 5 cm.

| ID | Terrain Type | Map Unit | Depth m | EOH material | Rock m | Longitude WGS84 | Latitude WGS84 | Elev. m | Date 2015 | Thaw depth m |
|------|---------|-----|------|----------|-------|-----------|----------|-----|--------|-------|
| 2008 | Eskers | GFr | 6.2 | ice poor | auger | -110.1851 | 64.6003 | 458 | 12 Jul | – |
| 2028 | Eskers | GFr | 11.7 | thawed | auger | -110.1851 | 64.6003 | 458 | 19 Jul | – |
| 2029 | Eskers | GFr | 12.1 | thawed | auger | -110.1846 | 64.6002 | 458 | 20 Jul | – |
| 2026 | Eskers | GFr | 1.5 | boulder | auger | -110.3451 | 64.7246 | 437 | 19 Jul | – |
| 1006 | Organic | Th | 4.4 | ice rich | 0.20 | -110.2333 | 64.6037 | 443 | 12 Jul | 0.35 |
| 1005 | Organic | Tv | 4.9 | ice | 0.00 | -110.2356 | 64.5999 | 440 | 10 Jul | 0.35 |
| 2009 | Till | Th | 5.0 | ice | 0.40 | -110.2169 | 64.6041 | 446 | 12 Jul | >0.34 |
| 2033 | Till | Th | 5.1 | ice rich | 0.45 | -110.2152 | 64.6055 | 458 | 22 Jul | >0.40 |
| 1004 | Till | Th | 4.3 | boulder | 0.75 | -110.2381 | 64.5951 | 471 | 9 Jul | – |
| 1007 | Till | Th | 4.3 | boulder | 0.25 | -110.2328 | 64.6037 | 442 | 12 Jul | 0.35 |
| 2004 | Till | Th | 6.0 | bedrock | 0.25 | -110.2363 | 64.5963 | 480 | 10 Jul | – |
| 2005 | Till | Th | 4.2 | ice rich | 0.05 | -110.2332 | 64.5966 | 472 | 11 Jul | >0.40 |
| 2006 | Till | Th | 4.3 | ice rich | 0.05 | -110.2307 | 64.5968 | 453 | 11 Jul | >0.47 |
| 2007 | Till | Th | 3.7 | boulder | 0.65 | -110.2354 | 64.5985 | 459 | 11 Jul | >0.40 |
| 1014 | Till | Th | 9.5 | ice | 0.50 | -110.7354 | 64.6254 | 489 | 21 Jul | – |
| 2018 | Till | R2 | 3.0 | bedrock | 0.00 | -110.4360 | 64.7136 | 461 | 16 Jul | 0.40 |
| 1009 | Valley | Tb | 5.3 | thawed | 0.10 | -110.4430 | 64.7015 | 432 | 14 Jul | 0.36 |
| 1010 | Valley | Tb | 5.0 | thawed | 0.00 | -110.4398 | 64.7027 | 426 | 14 Jul | 0.34 |
| 2011 | Valley | Tb | 5.8 | bedrock | 0.00 | -110.4479 | 64.7023 | 432 | 13 Jul | 0.27 |
| 2012 | Valley | O | 4.1 | bedrock | 0.00 | -110.4459 | 64.7021 | 433 | 13 Jul | >0.35 |
| 2013 | Valley | Tb | 3.3 | bedrock | 0.15 | -110.4501 | 64.7028 | 440 | 14 Jul | >0.40 |
| 2015 | Valley | O | 2.8 | thawed | 0.05 | -110.4421 | 64.7035 | 425 | 14 Jul | >0.40 |
| 2016 | Valley | Tb | 2.0 | boulder | 0.00 | -110.4441 | 64.7045 | 431 | 14 Jul | >0.40 |
| 2019 | Valley | R2 | 2.8 | ice rich | 0.70 | -110.4357 | 64.7021 | 428 | 16 Jul | 0.40 |

## Appendix B: Typical core

Figures B1, B2, and B3 show typical core.

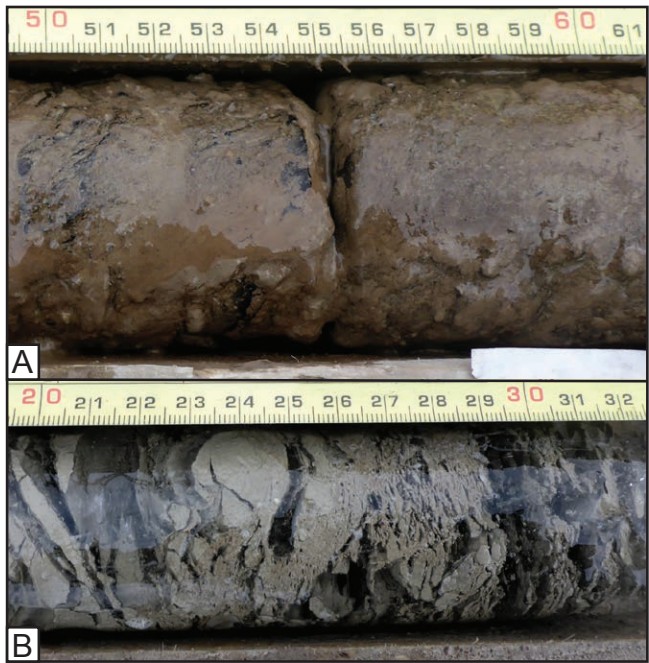

**Figure B1.** Drill core recovered from borehole NGO-DD15-1010 in the Valley. (A) shows core from 1.6 m depth with 46 % excess ice and 1.3 % organic matter. (B) is about 3.5 m depth with 60 % excess ice and 1.5 % organic matter. Scale bars are in cm.

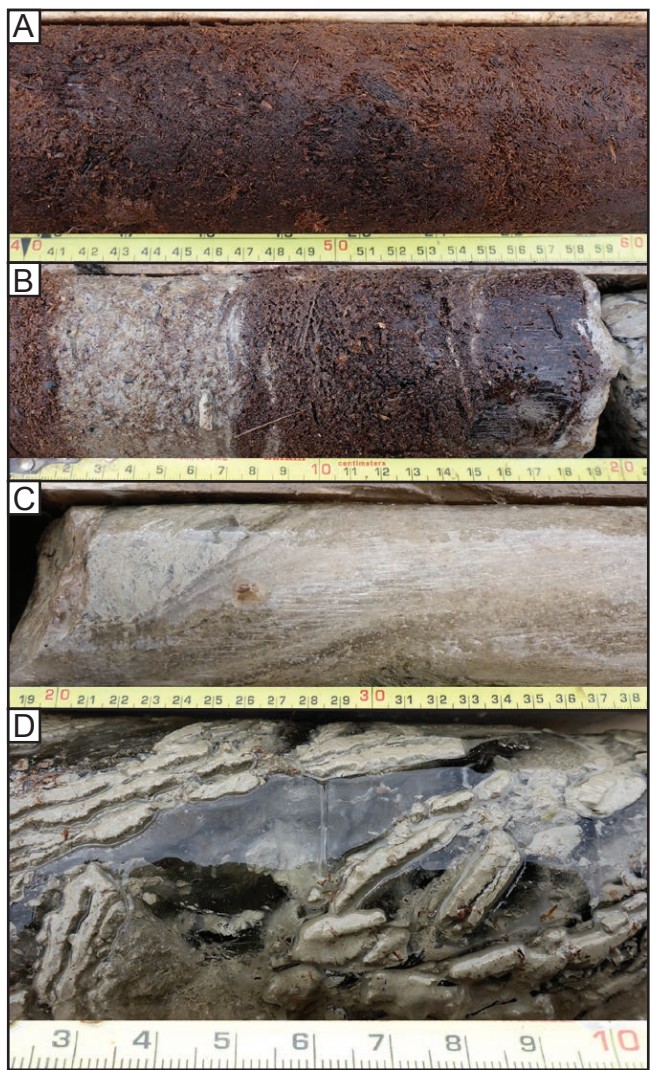

**Figure B2.** Drill core recovered from borehole NGO-DD15-1005 in organic terrain. (A) Compact frozen organics near 0.85 m without excess ice and 80 % organic matter. (B) Alternating layers of organics and mineral soil near 1.6 m with 19 % excess ice and 9.2 % organic matter. (C) Nearly pure ice around 2.7 m, no analyses available. (D) Ice in mineral soil, 3.5 m deep with 37 % excess ice and 1.6 % organic matter. Scale bars are in cm.

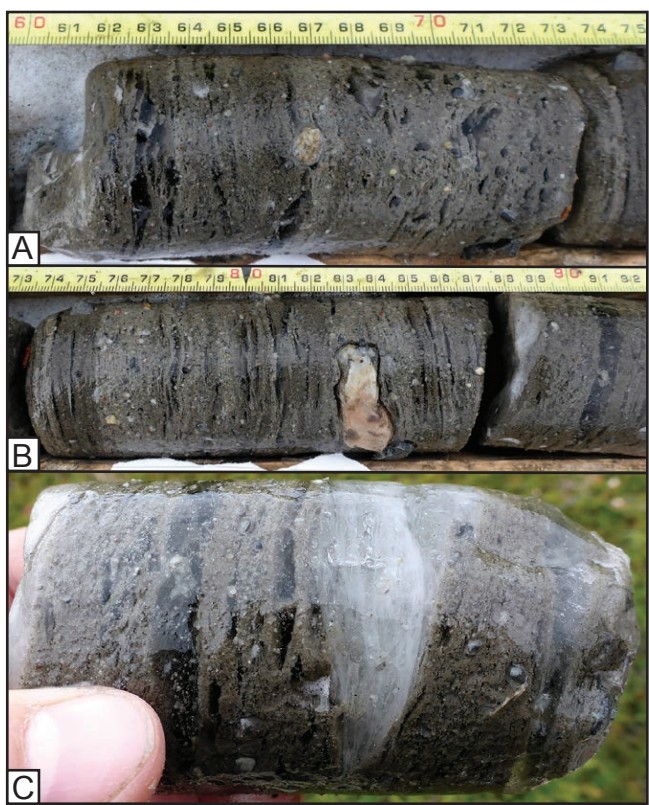

**Figure B3.** Drill core recovered from borehole NGO-DD15-2005 in hummocky till near a hilltop and logged as 40 % visible ice. (A) Core near 3.8 m depth with 43 % excess ice and 1 % organic matter. (B) Core near 4.0 m depth with 45 % excess ice and 0.6 % organic matter. (C) Closeup of the right side of the core shown in (B). Scale bars are in cm.

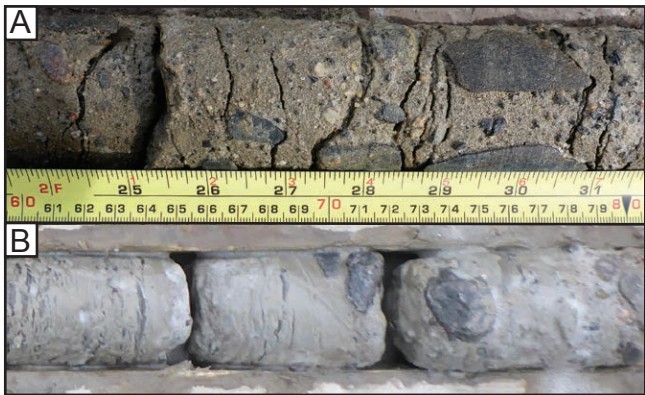

**Figure B4.** Evidence of permafrost in previously thawed till. (A) Thin lenses of segregated ice at 3.3–3.5 m in NGO-DD15-1007 indicate refreezing of thawed till and illustrate the different appearance when compared to the cryostructure in Fig. B3. (B) The contact between the well sorted silt/sand and coarser material a6 3.7–4.1 m in NGO-DD15-2012.

## Appendix C: Geomorphic evidence of post-glacial melt of ground ice

Figure C1 shows evidence of post-glacial ground-ice melt.

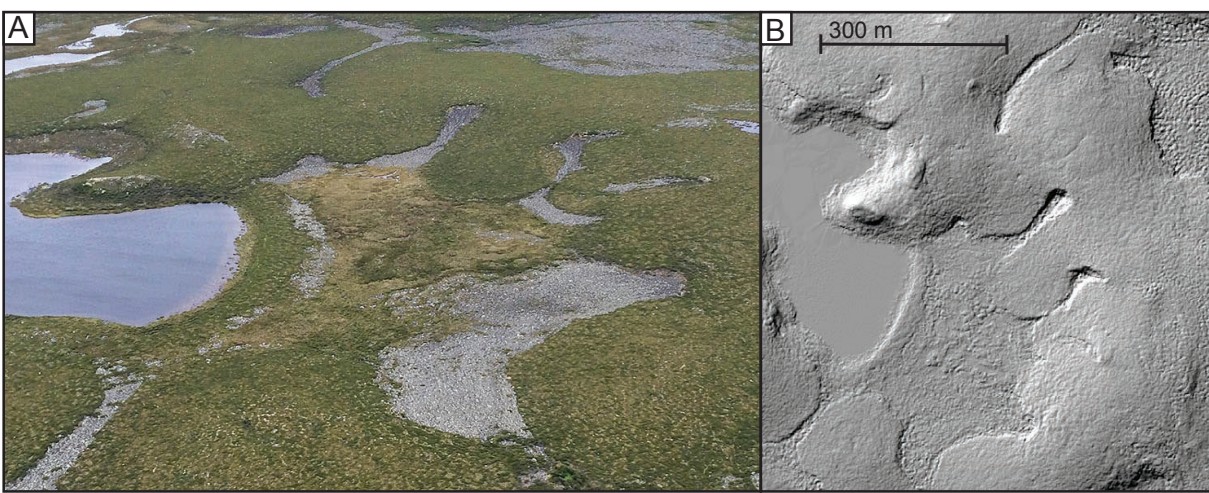

**Figure C1.** Terrain features that may indicate post-glacial ground-ice melt (see also McWade et al., 2017) shown in oblique aerial photograph from July 13, 2015 (A) and hillshade image with 1 m resolution (B). Lower elevation areas expose many large boulders. The elevation difference between the concave features and the smooth upland tills is 5–10 m and hypothesized to derive largely from loss of ice in the ground. The centre of the area shown is 109.944° W, 64.615° N. The hillshade is based on a digital elevation model kindly provided by Dominion Diamond Mines.

## Appendix D: Tabulated comparison with previous studies

Table D1 compares soil organic-carbon densities and Table D2 soluble cation concentrations between this and previous studies.

**Table D1.** Soil organic-carbon density (SOCd, $kg\ C\ m^{-3}$) per depth interval for three terrain types from the Lac de Gras study area (Table 2) compared with circumpolar mean values for similar soils reported in a recent compilation (Hugelius et al., 2014, Table 2). Upland Tills are compared to Turbels (cryoturbated permafrost soils), Eskers with Orthels (mineral permafrost soils unaffected by cryoturbation) and Organics with Histels (organic permafrost soils). Circumpolar values below 1 m are for "Thin sediment". For Orthels, values in "Thick sediment" are more than ten times larger.

| Depth | Histels | Organics | Turbels | Upland Till | Orthels | Eskers |
|---|---|---|---|---|---|---|
| 0–0.3 m | 60.3±10.0 | 26.5 | 49.0±5.0 | 14.8 | 52.7±8.7 | 17.8 |
| 0–1 m | 49.3±8.4 | 28.9 | 33.0±3.5 | 12.1 | 25.3±4.1 | 11.4 |
| 1–2 m | 49.0±9.2 | 25.6 | 31.6±33.1 | 7.8 | 2.6±2.0 | 3.7 |
| 2–3 m | 30.5±8.8 | 11.8 | 19.5±17.5 | 5.4 | 1.3±16.9 | 1.8 |

**Table D2.** Concentration of soluble cations in active layer and permafrost in mineral soils compared with previous studies in northwestern Canada that employ a comparable analytical approach. In this study, active-layer values derive from pit samples and permafrost values from frozen core sections, samples below 2 m depth were used on Eskers. Values from Lacelle et al. (2019) were derived using three sequential extractions in a 1:10 soil water ratio on dried soils, likely yielding higher concentrations (cf. Toner et al., 2013), possibly by a factor of two or more, than what would be obtained with the method described in this study.

| | Total soluble cations (meq/100 g dry soil) | |
|---|---|---|
| Terrain type | Active layer | Permafrost |
| *Lac de Gras area (this study), mean (median, n)* | | |
| Upland till | 0.06 (0.06, 18) | 0.24 (0.20, 59) |
| The Valley | 0.19 (0.10, 6) | 0.59 (0.29, 79) |
| Eskers | 0.05 (0.04, 18) | 0.15 (0.09, 10) |
| *Mackenzie Delta (Kokelj and Burn, 2005), mean values* | | |
| Point-bar willow | 2.06 | 3.83 |
| Point-bar alder | 1.54 | 2.15 |
| Spruce-alder-bearberry | 0.64 | 2.09 |
| Spruce-feathermoss | 0.49 | 1.24 |
| Spruce-crowberry-lich. | 0.38 | 1.62 |
| *Herschel Island (Kokelj et al., 2002), estimated from figures* | | |
| Undisturbed plateau | 0.25–1.5 | 12–14 |
| Undisturbed plateau | 0.25–6 | 8–14 |
| Undisturbed stable sl. | <0.25 | 5 |
| Disturbed AL detachm. | 2–10 | 12 |
| Disturbed AL detachm. | 2–12 | 12 |
| *Richardson Mountains–Peel Plateau, mean (median, n) (Lacelle et al., 2019)* | | |
| CB thaw slump | 1.21 (1.2, 12) | 7.0 (6.5, 12) |
| paleo AL | | 4.0 (3.8, 5) |
| Wilson slump | 2.12 (1.7, 5) | 6.4 (6.0, 6) |
| paleo AL | | 4.5 (4.1, 10) |
| WR-03 slump | | 10.4 (9.6, 9) |
| paleo AL | | 17.5 (15.4, 6) |
| WR-05 slump | | 11.6 (10.9, 7) |
| paleo AL | | 19.9 (21, 23) |

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
