# Peer review of "Ground ice, organic carbon, and soluble cations in tundra permafrost soils and sediments near a Laurentide ice divide in the Slave Geological Province, N.W.T., Canada"

_The Cryosphere, 2020_

## Referee Comment (RC1) · Anonymous Referee #1 · 6 Mar 2020

General comments

This study presents unique and highly valuable data on ground ice, organic carbon and soluble cation contents in deep permafrost cores of the Central Canadian Arctic. The surprisingly high ground ice content found in some cores makes the landscape susceptible to potential (differential) ground subsidence and thermokarst formation allowing the remobilization of deep carbon and soluble cation stocks (as well as affecting infrastructure).

[Figure]

The authors should better explain why this particularly study area (Lac de Gras) was selected, as well as which approach was used to select specific core sites. This is important information to evaluate how representative this study is for the wider Slave Geological Province.

There are some issues with field and laboratory procedures, regarding the logging of field volumes collected in the active layer of soil pits, the application of a mean LOI value of 80% to organic samples in the top meter of the cores, disregarding coarse clast volume, the inferred zero organic carbon content of the soil fraction 0.5-5.0 mm, the indirectly inferred fine soil fraction (< 0.5 mm) for about half of the samples, and the indirect derivation (regression) of dry bulk density values (when known volume samples are, or could have been, available for most of the samples). Particularly, SOC estimates for the 0-1 m depth interval are prone to large uncertainties and should not be the focus of the analysis. I feel it necessary to mention these concerns, even though in most cases they cannot be addressed any longer.

The structure and use of language are adequate. I propose to move one subsection on field sampling to Methods. Figures and tables are generally fine, but I suggest to add an additional map to Figure 1 as well as a new figure in the Appendix with properties of a few selected individual permafrost cores.

Despite some methodological issues, this study is a very important scientific contribution that addresses important gaps in the knowledge of ground ice and organic carbon content in deep permafrost cores (other than deltaic and Yedoma deposits).

Specific comments

Title: . . . and soluble cations in deep tundra permafrost cores near a Laurentide . . . Note: the 0-1 m (and active layer) SOC estimate is highly uncertain (see below), the authors should focus on the valuable deep data

Page 1 (P1), Line 13-14 (L13-14): . . . and 0-3 m, respectively. Deeper deposits have

[Figure]

C densities ranging from X-Y Kg C m-3, representing a significant additional C pool.

P1, L16: . . . and slightly less 0-3 m organic carbon stocks and fewer . . .

P2, L45: . . . consequences of permafrost thaw.

P2; L47: (Hugelius et al., 2014)

P3, L79: The authors should clarify why this particular study area was selected, especially since they compare their results to more generalized maps of permafrost/ground ice conditions and SOC storage for Canada and the northern circumpolar region. Is it a simple issue of accessibility, or was this area chosen because of special features of potential interest to infrastructure development (e.g., the occurrence of fossil thaw slumps as depicted in Fig. C1) ?. This is important in order to evaluate the representativity of the study area for the Slave Geological Province.

P3, L70: . . . characterization of active layer and deep permafrost materials in . . .

P3, L82: . . . and 14.x(0?) C, respectively, and . . .

P3, L83: cal yr BP ? (2 times)

P4, L95: I propose that the authors include a (simplified) surficial geological map of the 50x50 km study area as Fig. 1B, with location of the 24 permafrost cores. The current Canada map can be a small inset (Fig. 1A).

P4, L100: Organic soils cover 5% and . . .

P5, Fig. 2 caption (and related references in text). Shift B and C, see Figures 3-6

P5, L116: As with the study area, the authors should explain their selection of core sites. Were sites selected because of easy access, or because they were considered typical for the different surface geology units, or was there a degree of randomness in site selection. This is important to assess how representative sites are for scaling to the study area as a whole. See my point P8, L183-184.

P5, L116: Permafrost cores with a diameter of 5 cm were obtained in mid-July 2015 using a . . . . . . Note: So, these samples had a known volume that could have been used for DBD calculations !

P5, L118: How was the active layer in soil pits sampled ? Sample depth interval ?, using fixed volume cylinders (for DBD) ?

P6, L129: The coarse clasts >5 cm that could not be recovered by the drill are not considered. The authors refer to this on P12, L238. This could result in a significant over-estimation of OC densities, particularly in till. I wonder, are there no natural/excavated deep exposures in the general study area from which the volume proportion of large clasts can be (visually) estimated and then computed ?

P6, L149: Using a LOI of 80% for those samples with no visible mineral component is highly questionable. Peat deposits will normally have a higher LOI, whereas topsoil organics in mineral soils will have generally a lower LOI. This introduces high uncertainty, which is one reason why the authors should not focus on the SOC 0-1 m stock.

P6, L150: The LOI applies to the fine soil fraction (<0.5 mm), whereas the volume of the coarse fraction >5 mm – 5 cm is accounted for (P6, L129). But what happens to the fraction 0.5-5 mm, is this all considered 100% mineral ? It could include roots, or other plant remains / organic aggregates, etc. ?. Furthermore, the fine soil fraction (<0.5 mm) is only available for half of the samples and very indirect approaches are used to calculate this value for the remaining samples (P7, L166-167).

P7, L160: It is rather unfortunate that DBD was not computed directly from dry weight and field volume of samples, at least for those samples in which no ground ice/materials were lost

P7, L174-175: The fine fraction and DBD deviations for calculating uncertainty ranges seem to be quite arbitrary

P8, L183-184: This section/subsection should be moved prior to subsection 3.1., starting with an explanation about the selection of sites (see my point P5, L116)

P8, L187+: For all boreholes, it should be indicated why coring was discontinued (hitting bedrock, logistical/time constraints, etc.). See also comment on Table A1 (below)

P8, L218: Please add area proportion for each surface geology class (see P15, L316-317)

P9-12: In Figs. 3-6 the authors have grouped samples from all profiles belonging to one class in one and the same graph. The information from single profiles is lost. I propose to add graphs in the Appendix, providing data from Figs. 4-6 for a single/typical/most complete core for each surface geology class (New Appendix Figure C1-C4). It should be considered that data from individual profiles are more valuable than composites that cannot be disentangled anymore in its individual components.

P13, L264-265 and P14, L274-275: (currently Fig. C1)

P18, Table A1: Add depth of core (and reason to stop drilling)

P20, Fig. B2 caption. The peat in (A) would normally have a LOI of c. 95%. The default value of 80% does not generally apply

---

## Short Comment (SC1) · 27 Mar 2020

Comments to tc-2020-33 Discussion Paper by Subedi, Kokelj and Gruber

Stephen Wolfe - Geological Survey of Canada

Introduction

The paper under review by Subedi, Kokelj and Gruber provides data on ground ice, organic carbon, and soluble cations from drill holes in the Lac de Gras region of the

[Figure]

Slave Geological Province, Canada. The authors indicate that their results differ from observations made in western Arctic Canada and they make specific comparisons to field studies within the Mackenzie Delta region. They further indicate that the study provides quantitative data for a region that has few previous studies.

However, several statements and conclusions by the authors require reappraisal and revision in light of existing literature. Three issues in particular merit attention. Regarding the glacial context, the authors contend that the site is near the Keewatin Ice Divide, although it is more than 500 km distant. In reporting ground ice, they interpret model outputs prepared at a national scale in a local context, and combine surficial units with critically different properties. Lastly, the authors overlook relevant studies published from the Slave Geological Province, including from the study area.

Literature from the Slave Geological Province is discussed here to assist the authors in their task.

Glacial Context

The authors represent the study site as near a Laurentide Ice divide and having been influenced directly by it when, in fact, the Keewatin Ice Divide is distal to the study area.

The title of the paper indicates that the study area is "near a Laurentide ice divide", while line 29 states that the Lac de Gras region "is situated close to the Keewatin Ice divide of the Laurentide Ice Sheet". The term "Laurentide ice divide" should not be used. "Laurentide Ice sheet ice divide" or" Keewatin Sector ice divide" are appropriate alternatives. Laurentide Ice Sheet ice divides comprise named ice divides (with capital letters: Keewatin Ice Divide, Labrador Ice Divide, M'Clintock Ice Divide, etc.) and unnamed ice divides (see Dyke and Prest, 1987). However, the Keewatin Ice Divide (KID) is by definition "the zone occupied by the last glacial remnants of the Laurentide Ice Sheet west of Hudson Bay" (Lee et al., 1957). Therefore, the KID was situated at least 500 km east of the study area (e.g. McMartin and Henderson, 2004). All the other ice divide positions in Keewatin are not located closer to the study area and cannot be

termed "Keewatin Ice Divide".

If the authors wish to consider features 500 km from their study area as proximal, then a reader would further expect literature that reports conditions in the glacial sediments near Yellowknife, only 310 km from the study area (line 79), to be fully considered. This is addressed further below.

Basal ice sheet conditions, as discussed by Rampton (2000) and Utting et al. (2004), influence the source materials for shield-derived tills of the Slave Geological Province. Glacial conditions in the Slave Geological Province differed significantly from those of western Arctic Canada. Such differences had a profound effect on ground ice development (Wolfe et al., 2017). The authors compare their results with conditions in western Arctic Canada, where the source materials for till were derived from the sedimentary basins of the Interior Plains as opposed to the Canadian Shield. It is not evident why this setting is the primary reference for comparison with results from this study without relevant details on conditions in the Slave Province.

Ground ice reporting

Use of model outputs

The authors apply national-scale modelling results to local site conditions.

The authors state that "the new Ground Ice Maps for Canada (O'Neill et al., 2019) show the study area (50 km × 50 km) to contain no or negligible wedge ice, negligible to low segregated ice and no relict ice, which includes buried glacier ice." (lines 40-42). In fact, the new Ground Ice Maps for Canada depict wedge ice from none to low; segregated ice from none to medium, and relict ice (which includes buried glacier ice) from none to low (Figure 1). The authors thereby under-report the amount of ground ice depicted for the study areas by O'Neill et al. (2019).

Nevertheless, differences between the authors' reporting and the model results are due, in part, to site-specific surficial geology. The surficial geology used in modelling

is at scale of 1:5 M (GSC - Surficial Geology of Canada, 2014). For associations between ground ice and surficial geology to be appropriately considered at the local scale of 1:125,000, Dredge et al. (1995) and Ward et al. (1997) may be consulted.

Use of surficial geological units

The authors combine surficial units with critically different properties in the context of reporting excess ice.

The authors combine drill cores into "upland tills", which they define as "smoothly rounded hills comprised of thick till and in till veneer over bedrock" (line 187-188). Combining drill cores from till veneers, which are tills that are less than 2 m in thickness, with the drill cores of thick till misrepresents the extent of "upland till" and therefore of ground ice contained within till terrain. To this end, Ward et al. (1997) and Dredge et al. (1995) provide more suitable spatial depiction of till veneer, till blankets, and hummocky till that permit the drill cores to be allocated to these specific till units. Such separation is appropriate for depicting depth versus water content and excess ice (as in Figure 3A and 4A). This approach may highlight the lower excess ice abundance in till veneers and at depths above 4 m, and higher amounts in thicker tills (and at depths below 5 m). These data may further inform understanding of the proportion of hummocky till, and thus potentially preserved Laurentide basal ice, in the area.

In addition, in presenting "fluvially reworked till (the Valley)" (line 185) where "silts and sands are well sorted and likely derived from fluvial reworking of local tills" (line 191) and in presenting evidence of post-glacial ground ice melt features (e.g. Figure C1) the authors might acknowledge alternative interpretations by Rampton (1999) and by Utting et al. (2009) to account for fluvial activity. Alternative classification of the terrain types is required because the current terms conflate categories of phenomena, e.g. till and valley.

Incorporation of comparative literature

[Figure]

The authors overlook existing regional and local literature with similar conclusions, thereby claiming undue precedence.

Solutes in mineral soils

The authors' state that "the concentration of total soluble cations in mineral soils is much lower than at other previously studied locations in the western Canadian Arctic" (lines 14-15) and that "the absolute concentrations of soluble cations obtained in the study area near Lac de Gras are low compared to previous studies from northwestern Canada that report higher concentrations in active layer and permafrost across diverse terrain types (Table D2)." (lines 302-304). These remarks assume that all comparable previous studies have taken place in the Mackenzie delta area or Herschel Island (Table D2). The authors indicate that "The low concentrations in our study area are associated with the contrasting nature and origins of surficial materials. Tills in our study region are generally coarser grained than many glacial deposits studied in the western Arctic, are regionally sourced from mostly granitic rocks and have been exposed only to minor postglacial landscape modification (Haiblen et al., 2018; Rampton and Sharpe, 2014)". (lines 311-314).

Gaanderse et al. (2018) originally reported on solute concentrations from glaciolacustrine deposits within the shield area that indicate low values similar to the Lac de Gras area. Gaanderse et al. (2018, p. 1039 noted that "Total soluble ions concentrations decreased with depth from the active layer to the underlying glaciolacustrine clays in permafrost (Figure 8). This trend contrasts with observations in the western Arctic, where low ion concentrations occur within sediments of the active layer and near-surface permafrost, relative to the underlying permafrost (Kokelj et al., 2002; Kokelj and Burn 2003, 2005; Lacelle et al., 2014). Unlike the predominantly marine origin and the mixed-layered clays of the western Arctic (Dewis et al., 1972; Kodama, 1979), the glaciolacustrine clays of the Great Slave region are not inherently solute-rich or weathered." And "These fine-grained glaciolacustrine, lacustrine and alluvial sediments belong to the same generation of glacially-derived sediments with a regional

**TCD**

[Figure]

mineralogical composition from igneous and metamorphic sources (Aden et al., 2015). The clays and clay-sized glaciolacustrine sediments are predominantly unweathered, with major soil ion abundances likely reflecting the mineralogy of local rocks, including $Ca_{2+}$, $Na_{2+}$, and $K_+$ from the weathering of feldspars; $Mg_{2+}$ and $Ca_{2+}$ from amphiboles, pyroxenes and olivine; $SO_{42-}$ from sulphides and $Cl_-$ from igneous sources."

Additional supporting data on low concentrations of soluble cations from the Slave Geological Province and in the Lac de Gras area are also presented in Wolfe et al. (1997a) and Wolfe (1998) who describe low cation concentrations in buried ground ice in glaciofluvial delta sediments, and in Wolfe et al. (1997b), who include borehole logs, geophysical surveys, cation concentrations and oxygen isotopes from a 14-m borehole at the BHP Airstrip Esker within the authors' Lac de Gras study area.

Ground ice expectations

The authors state that "permafrost in the study area contains much more ground ice than expected" (line 16). As noted above, several studies have illustrated the presence of high ground ice contents in outwash sediments in the area. In addition to these, Dredge et al. (1999), referenced by the authors, clearly present expected ground ice conditions and terrain sensitivity in line with the authors observations. The importance of the geological legacy in determining the characteristics of permafrost and potential responses of this system of disturbance and change is further summarized in Wolfe et al. (2017), who conclude that "Glacially-derived ground ice includes buried glacial ice within glaciofluvial outwash deposits and buried glacial and meltwater ice within eskers. Sediment-rich ice has also been encountered within hummocky till terrain during mine development operations. Surficial features attributed to partial thawing and creep of massive ground ice are regionally apparent. Although massive ice has been encountered in only a few locations to date, buried ground ice may be common within this glaciated region of the Tundra Shield."

Nevertheless, the authors are still cautioned about asserting that "Tills in our study
areas ... have been exposed to only minor post-glacial modification (Haiblen et al., 2018; Rampton and Sharpe, 2014)", noting the evidence of Holocene warming and tree-line advance in the region as noted by the authors (lines 83-84) and in Moser and MacDonald (1990) and MacDonald et al. (1993).

The authors state in the abstract that "thaw subsidence of metres to more than ten metres is possible" due to ground ice that may be buried Laurentide basal ice (line 8 – 9). Within the paper, the authors write: "A potential surface lowering of many metres, up to more than ten metres, is thus to be expected from areas of thick till if this permafrost was to thaw completely" (lines 271-272). Again, in the conclusions, the authors state: "Thaw-induced terrain subsidence on the order of metres to tens of metres is possible in ice-rich till" (line 326). The statements are based upon data from only one borehole with samples from below 6 m depth. The borehole terminated at 9.5 m depth. The authors assume that conditions in this borehole are representative of all till "estimated to be 10-30 m thick in the area (Haiblen et al. 2018)" (line 270). In other words, the authors assume, without disclosed evidence, that the excess ice profile presented in Fig. 4 extends indefinitely downwards with high values, that it applies consistently throughout an extensive till unit, and that the unit is sufficiently thick to contain excess ice tens of metres thick. Readers should be made aware of the assumptions upon which these statements are based and, in particular, should be able to recognize that the principal data contributing to these assertions are derived from 3.5 m of drill core, from 6 to 9.5 m in the profile.

Summary and Conclusion

The paper by Subedi, Kokelj and Gruber (in review) provides an added contribution to the growing knowledge of permafrost and ground ice in the Slave Geological Province. The purpose of these comments is to provide an appropriate regional context for the observations. The authors may take advantage of these comments so that their contribution to the literature will complement, and be informed by, the existing knowledge of permafrost, environmental change and ground ice conditions in this region.

Acknowledgements

These comments benefitted from discussion and input with several colleagues. In particular, Drs. Chris Burn, Brendan O' Neill, Peter Morse, Dan Kerr and Isabelle McMartin are gratefully acknowledged.

References:

Aden, A.A., Wolfe, S.A., Percival, J.B. and Grenier, A.: Characteristics of glacial Lake McConnell clay, Great Slave Lowland, Northwest Territories; Geological Survey of Canada, Current Research 2015-7, 12 p, 2015.

Dredge, L.A., Ward, B.C. and Kerr, D.E.: Geological Survey of Canada, "A" Series Map 1867A, 2 sheets https://doi.org./10.4095/207631, 1995.

Dyke, A. and Prest, V.: Late Wisconsinan and Holocene history of the Laurentide ice sheet. Géographie physique et Quaternaire, 41, 237-263, 1987.

Gaanderse, A.J.R., Wolfe, S.A., and Burn, C.R.: Composition and origin of a lithalsa related to lake‐level recession and Holocene terrestrial emergence, Northwest Territories, Canada. Earth Surface Processes and Landforms 43, 1032-1043, 2018.

Geological Survey of Canada.: Surficial Geology of Canada. Geological Survey of Canada, Canadian Geoscience Map 195, (ed. Prelim., Surficial Data Model V.2.0 Conversion), 1 sheet, https://doi.org/10.4095/295462, 2014.

Lee, H.A., Craig, B.G. and Fyles, J.G.: Keewatin Ice Divide. Geological Society of America, Bulletin 68, 1760-1761, 1957.

MacDonald, G.M., Edwards, T.W.D., Moser, K.A., Pientiz, R., and Smol, J.P.: Rapid response of treeline vegetation and lakes to past climate warming. Nature, 361, 243-246, 1993.

McMartin, I. and Henderson, P.J.: Evidence from Keewatin (central Nunavut) for paleo-ice divide migration. Géographie physique et Quaternaire, 58, 163-186, 2004.

Moser. K.A. and MacDonald, G.M. : Holocene vegetation change at treeline north of Yellowknife, Northwest Territories, Canada. Quaternary Research, 34, 227-239, 1990.

O'Neill, H.B., Wolfe, S.A. and Duchesne, C.: New ground ice maps for Canada using a paleogeographic modelling approach. Cryosphere 13, 753–773, 2019.

Rampton, V.N.: Large-scale effects of subglacial meltwater flow in the southern Slave Province, Northwest Territories, Canada Can. J. Earth Sci. 37, 81–93, 2000.

Utting, D.J., Ward, B.C. and Little, E.C.: Genesis of hummocks in glaciofluvial corridors near the Keewatin Ice Divide, Canada. Boreas, 38, 471-481, 2009.

Ward, B.C., Dedge, L.A. and Kerr, D.E.: Geological Survey of Canada, "A" Series Map 1870A, 2 sheets, https://doi.org/10.4095/209260, 1997.

Wolfe, S.A.: Massive ice associated with glaciolacustrine delta sediment, Slave Geological Province, N.W.T., Canada. In: Proceedings Seventh International Conference on Permafrost, Yellowknife, NWT. A.G. Lewkowicz and M. Allard (eds). Collection Nordicana, University Laval, 1998.

Wolfe, S.A., Kerr, D.E. and Morse, P.D. 2017.: Slave Geological Province: an archetype of glaciated shield terrain. In: Landscapes and landforms of western Canada. O. Slaymaker (Ed). World Geomorphological Landscapes. Springer International Publishing, Switzerland.

Wolfe, S.A., Burgess, M., Douma, M., Hyde, C., Robinson, S.: Geological and geophysical investigations of ground ice in glaciofluvial deposits, Slave Province, District of Mackenzie, Northwest Territories.; in Current Research 1997-C: Geological Survey of Canada, 39-50. 1997a.

Wolfe, S.A., Burgess, M.M., Douma, M., Hyde, C. and Robinson, S.: Geological and geophysical investigations of massive ground ice in glaciofluvial deposits, Slave Geological Province, Northwest Territories. Geological Survey of Canada, Open File 3442. 50 p. 1997b.

Additional references for consideration:

BHP Diamonds Inc.: NWT Diamonds Project environmental impact statement. Vol. 1-4. BHP Diamonds Inc., Vancouver, B.C., 1995.

EBA Engineering Consultants Limited: Koala mine airport esker evaluation. EBA Report No 0101-94-11439.3, submitted to BHP Diamonds Incorporated, Vancouver, B.C., 1995.

[Figure]

[Figure]

Borehole locations                    Wedge Ice

[Figure]

Segregated ice                        Relict ice

**Fig. 1.** Figure 1. Google Earth image of study area (pins are specific borehole locations from
the present study) and model output results from O'Neill et al. (2019).

---

## Referee Comment (RC2) · Anonymous Referee #2 · 9 Apr 2020

Review: tc-2020-33

Summary

This paper is formally well written and presented, and describes unique data from an interesting environment that is under-represented in the literature. I think that the data from these cores should be published, but in its current form the paper does not do the material justice. The paper suffers from a mismatch between its stated goals and its delivered conclusions. There are also problems with the methods that are major

enough to call some of the conclusions into question. I recommend that the paper be rejected and returned to the authors for re-working and potential re-submission after the problems have been addressed.

Criticisms

Approach

The context for this study requires more thinking and description. For example, the authors state that, contrary to published maps, there is significant excess ice in the "area" – what is the area? The paper requires a map showing the Keewatin Ice Divide, the "central Slave Geological Province", etc. Table A.1 would be informative if it were a map or, even better, mapped together with an optical satellite image to allow comparison with land cover. For what region do the authors believe that their results are representative? Demonstrate that the set of cores is somehow representative for a region and communicate what region this is. How much of the region that you so identify is represented by the four terrain types for which you present data?

"Comparison with other permafrost regions . . ." is listed as the 3rd goal of the paper, but is not addressed and represents a central weakness. As an example, any reader of this paper would expect to find permafrost ONLY in those regions of the NWT and Yukon in which the authors have already worked, as well as in unglaciated portions of Alaska. Such myopia is not unique: many authors cite only papers within their national borders, but this criticism is a grave one for geoscientists, who should have a natural curiosity for the variability of expressions of their study objects on Earth. European, Alaskan and Russian research on permafrost is extensive, and offers many studies that deal with ALL of the study objects (glaciation, permafrost, ground ice, ground ice/water chemistry, etc.) and processes (thawing, landscape change, deglaciation, solute exclusion, thermokarst, etc.) in this study. Willfully ignoring most of the research in your field makes it impossible for the authors to demonstrate rigor in their approach, their knowledge of their field or an openness to the existing and potentially alternative interpretations of similar data sets. Comparing your results (e.g. Table D2) to only 2 previous studies (BOTH from your own research group!) cannot fulfill your stated goal to "...compare excess-ice content, organic-carbon density and soluble cation concentration with other permafrost environments...". It certainly reduces the relevance of your work to a broader readership and would alone be reason enough not to publish this manuscript.

The study makes the a priori assumption that categorization of the soil cores by terrain types is justified and that averaging all borehole results within one terrain type is justififed. How were terrain types identified? Why were these four chosen? Why do the authors think that terrain type has an influence on ice content, organic carbon, and total soluble cations? Such questions are fundamental to study design and the means to reaching conclusions of broader significance. Tellingly, there is no significance to the terrain types in the 6 conclusions reached. At this point, the authors should have re-examined the basis for their choices. The results for all four parameters as a function of depth do not provide enough information (Fig. 3-6) for the reader to assess whether the cores really differ between terrain types. It is not clear why only total soluble cations are analyzed and presented. Why not other dissolved species? Why not present profiles for individual boreholes and/or cations? This work is left to the reader.

Methods & Data treatment

The field photos of the cores are very impressive and show a wide variety of compositions and cryostructures – I think the paper would be strengthened by more of these and closer examination of the results.

The methods used in this paper are beset with problems. The authors should refer to standard texts and guides on soil analyses for methods of analyses. It is not sufficient to cite previous studies by the same set of authors (using "cf."?) to establish that standard accepted methods have been applied. Specific problems with the analytical methods are:

- The are 2 sets of cation concentrations: those which underwent an arbitrary 1:1 dilution with water and those which did not. Diluting the solution affects the equilibrium between dissolved and adsorbed species. It is not acceptable to treat these data sets as equivalent expressions of some kind of concentration without at least testing for equivalence.

- The creation of means as a function of depth is problematic and has not been justified – how is the 95% confidence limit calculated? In many cases MOST of the measurements lie outside of the 95% confidence limit. In what are the authors 95% confident? Anyone drilling in a similar location can be confident that most values lie outside of these limtis. With this kind of variability over depth, drawing a smoothed mean variation (the blue lines) over depth is nonsensical and masks real variability. It certainly requires examining each set of measurements on a per-borehole basis and on evaluating the data in detail.

- Similarly for Table D2: it is not clear how many samples are used to form the means, whether they are means, what the variability is, etc.

- Presumably data from multiple cores is combined for each of the four regions – it is not clear and has not been established that these groups of cores are similar enough to be grouped, that the cores cover similar depth ranges, that the sampling frequency is similar, or that the sample sizes per terrain type have no effect. Present the data for individual cores, do not create means, etc. and then establish that the groups of cores are different using a test of significance. At the moment, all the work is left to the reader.

- The expression of concentration per dry weight of sediment is almost entirely meaningless. It is meaningless in terms of processes affecting concentration during freezing, thawing or in general in the field. Concentration gradients, moisture migration, and any other relevant processes will depend on concentrations in the pore space or pore water or liquid water. Concentrations should AT LEAST be reported in terms of the water

volume obtained by thawing the samples.

- The calculation of excess ice content based on the ratio of volumes of thawed, saturated, settled sediment and supernatant liquid is problematic since the volume of supernatant liquid depends on soil texture.

Conclusions

Each conclusion has problems:

1. Without placing your borehole sites in a geographical context, it is difficult to evaluate whether this qualifies as a new regional insight.

2. The method of measuring excess ice does not allow the conclusion that thick occurrences of excess ice were found in tills. The photos show excess ice, but make it difficult to believe the volumetric values obtained by this method.

3. The soil cores go down to ten metres and have maximal ice contents of 60%. If the deepest core had the highest ice content, you would have subsidence of less 6 m. How then is subsidence of "tens of metres" possible? Is this based on some kind of unmentioned extrapolation of observations?

4. These are potentially interesting values, but would be made relevant if there was someway to know for what region the authors claim they are representative.

5. The cation concentration data are used only to establish in general "lower concentrations" when compared to a two studies from one other region and are entirely incidental to the paper's conclusions. There is no need to present terrain types, variation over depth or any of the data to reach this conclusion.

6. I agree that geological legacy is important. The data here are insufficiently linked to geological legacy.

---

## Author Comment (AC1) · 29 Jun 2020

Reply to the interactive comment of Anonymous Referee 1 concerning the manuscript "Ground ice, organic carbon and soluble cations in tundra permafrost and active-layer soils near a Laurentide ice divide in the Slave Geological Province, N.W.T., Canada" by R.Subedi, S.V. Kokelj and S. Gruber.

We are grateful for the constructive and detailed comments provided. Here, we respond to each issue raised and outline how the comments led us to revise the

manuscript. The entire original text of the interactive comment is shown in **bold font** and author responses in regular font. Each issue is identified with a code indicating "Referee 1" as well as a consecutive number, e.g., R1.1. This will help to revisit key issues raised by each of the three interactive comments in a summary reply that outlines the most important changes to the manuscript.

**0.1 General comments**

**This study presents unique and highly valuable data on ground ice, organic carbon and soluble cation contents in deep permafrost cores of the Central Canadian Arctic. The surprisingly high ground ice content found in some cores makes the landscape susceptible to potential (differential) ground subsidence and thermokarst formation allowing the remobilization of deep carbon and soluble cation stocks (as well as affecting infrastructure).**

**The authors should better explain why this particularly study area (Lac de Gras) was selected, as well as which approach was used to select specific core sites. This is important information to evaluate how representative this study is for the wider Slave Geological Province.**

**There are some issues with field and laboratory procedures, regarding the logging of field volumes collected in the active layer of soil pits, the application of a mean LOI value of 80% to organic samples in the top meter of the cores, disregarding coarse clast volume, the inferred zero organic carbon content of the soil fraction 0.5-5.0 mm, the indirectly inferred fine soil fraction ($<$ 0.5 mm) for about half of the samples, and the indirect derivation (regression) of dry bulk density values (when known volume samples are, or could have been, available for most of the samples). Particularly, SOC estimates for the 0-1 m depth interval are prone to large uncertainties and should not be the focus of the analysis. I feel it necessary to mention these concerns, even though in most cases they**

**cannot be addressed any longer.**

**The structure and use of language are adequate. I propose to move one sub-section on field sampling to Methods. Figures and tables are generally fine, but I suggest to add an additional map to Figure 1 as well as a new figure in the Appendix with properties of a few selected individual permafrost cores.**

**Despite some methodological issues, this study is a very important scientific contribution that addresses important gaps in the knowledge of ground ice and organic carbon content in deep permafrost cores (other than deltaic and Yedoma deposits).**

0.2   Specific comments

**Title:...and soluble cations in deep tundra permafrost cores near a Laurentide... Note: the 0-1 m (and active layer) SOC estimate is highly uncertain (see below), the authors should focus on the valuable deep data**

*R1.1 – Deep cores:* Interpretation of the term 'deep' varies between academic communities and, as such, we prefer to not use it in the title. The abstract clarifies "Twenty-four boreholes with depths up to ten metres...".

**Page 1 (P1), Line 13-14 (L13-14): . . . and 0-3 m, respectively. Deeper deposits have C densities ranging from X-Y Kg C m-3, representing a significant additional C pool.**
Done

**P1, L16: ...and slightly less 0-3 m organic carbon stocks and fewer...**
Sentence omitted, now.

**P2, L45: ...consequences of permafrost thaw.**
Done

**P2; L47: (Hugelius et al., 2014)**
Done

**P3, L79: The authors should clarify why this particular study area was selected, especially since they compare their results to more generalized maps of permafrost/ground ice conditions and SOC storage for Canada and the northern circumpolar region. Is it a simple issue of accessibility, or was this area chosen because of special features of potential interest to infrastructure development (e.g., the occurrence of fossil thaw slumps as depicted in Fig. C1) ?. This is important in order to evaluate the representativity of the study area for the Slave Geological Province.**

*R1.2 – Selection of study site:* More detail on the selection of the study site and its relationship with surrounding areas is now provided, including new map figures.

**P3, L70: ...characterization of active layer and deep permafrost materials in...**
We prefer to avoid the term 'deep permafrost', see R1.1.

**P3, L82: ...and 14 C, respectively, and...**
Done

**P3, L83: cal yr BP ? (2 times)**

*R1.3 – Calibrated years:* The two references cited do not specify whether these are calibrated radiocarbon years. Presumably they are, but we prefer not to make that determination.

**P4, L95: I propose that the authors include a (simplified) surficial geological map of the 50x50 km study area as Fig. 1B, with location of the 24 permafrost cores. The current Canada map can be a small inset (Fig. 1A).**

*R1.4 – Map of study area:* This figure is now provided.

**P4, L100: Organic soils cover 5% and...**
This has now been replaced by table because the response to other comments required more specific data on spatial abundances.

**P5, Fig. 2 caption (and related references in text). Shift B and C, see Figures 3-6**
Done

**P5, L116: As with the study area, the authors should explain their selection of core sites. Were sites selected because of easy access, or because they were considered typical for the different surface geology units, or was there a degree of randomness in site selection. This is important to assess how representative sites are for scaling to the study area as a whole. See my point P8, L183-184.**

*R1.5 – Site selection and sampling strategy:* The original strategy for site selection is now explained in more detail.

**P5, L116: Permafrost cores with a diameter of 5 cm were obtained in mid-July 2015 using a ... Note: So, these samples had a known volume that could have been used for DBD calculations !**

*R1.6 – Volume of core:* Many of the recovered core sections were irregular due to reaming or partial melting. This is now clarified in more detail and with an additional

figure in the revised text.

**P5, L118: How was the active layer in soil pits sampled ? Sample depth interval?, using fixed volume cylinders (for DBD) ?**

*R1.7 – Volume of pit samples:* Pits were sampled, where possible, at depths of 10 cm, 20 cm and 30 cm. The use of sampling cylinders deviated from the intended protocol and the volume cannot be reconstructed with confidence. Clarified in revised text.

**P6, L129: The coarse clasts >5 cm that could not be recovered by the drill are not considered. The authors refer to this on P12, L238. This could result in a significant overestimation of OC densities, particularly in till. I wonder, are there no natural/excavated deep exposures in the general study area from which the volume proportion of large clasts can be (visually) estimated and then computed?**

*R1.8 – Bias from clasts larger than 5 cm* The average volume of clasts coarser than the drill barrel has now been estimated visually from the core photographs and applied as a correction in the aggregation of organic carbon densities and storage, accordingly.

**P6, L149: Using a LOI of 80% for those samples with no visible mineral component is highly questionable. Peat deposits will normally have a higher LOI, whereas topsoil organics in mineral soils will have generally a lower LOI. This introduces high uncertainty, which is one reason why the authors should not focus on the SOC 0-1 m stock.**

*R1.9 – Uncertainty from LOI estimate when no mineral content visible:* Yes. This now stated and referenced in the revised manuscript. Estimates actually only affected the

pit samples in the top 0.3 m and this is also clarified, now.

**P6, L150: The LOI applies to the fine soil fraction ($<$0.5 mm), whereas the volume of the coarse fraction $>$5 mm – 5 cm is accounted for (P6, L129). But what happens to the fraction 0.5–5 mm, is this all considered 100% mineral ? It could include roots, or other plant remains / organic aggregates, etc. ?. Furthermore, the fine soil fraction ($<$0.5 mm) is only available for half of the samples and very indirect approaches are used to calculate this value for the remaining samples (P7, L166–167).**

*R1.10 – Carbon density bias from 0.5–5 mm grain-size fraction:* After drying the samples at 105 C, they were crushed with mortar and pestle before sieving; most root or plants residues would have been crushed and passed the sieve.

**P7, L160: It is rather unfortunate that DBD was not computed directly from dry weight and field volume of samples, at least for those samples in which no ground ice/materials were lost**
Agreed. At the same time, this is a unique set of data and we intend to make it as useful as is possible.

**P7, L174-175: The fine fraction and DBD deviations for calculating uncertainty ranges seem to be quite arbitrary**

*R1.11 – Arbitrary uncertainty ranges* They are based on reasoning and subjective decisions because no objective values can be determined. This is why we wrote 'The potential magnitude of this effect...' rather than calling it an estimation of uncertainty. More explanation has now been added in the revised manuscript to make the reasoning behind those ranges traceable, and therefor the resulting values more valuable.

**P8, L183-184: This section/subsection should be moved prior to subsection 3.1., starting with an explanation about the selection of sites (see my point P5, L116)**

*R1.12 – Moving Section 4.1 Study Sites* We have expanded the section on field observation and sampling to also describe the original sampling protocol and difficulties encountered.

**P8, L187+: For all boreholes, it should be indicated why coring was discontinued (hitting bedrock, logistical/time constraints, etc.). See also comment on Table A1 (below)**

*R1.13 – Expanding Table A1:* The table has been expanded and the Supplement now contains plots per borehole.

**P8, L218: Please add area proportion for each surface geology class (see P15, L316-317)**

*R1.14 – Abundance of surficial geology classes:* This is now in a new table.

**P9-12: In Figs. 3-6 the authors have grouped samples from all profiles belonging to one class in one and the same graph. The information from single profiles is lost. I propose to add graphs in the Appendix, providing data from Figs. 4-6 for a single/typical/most complete core for each surface geology class (New Appendix Figure C1-C4). It should be considered that data from individual profiles are more valuable than composites that cannot be disentangled anymore in its individual components.**

*R1.15 – Individual borehole profiles:* This has been added as supplementary material.

**P13, L264-265 and P14, L274-275: (currently Fig. C1)**
Fixed

**P18, Table A1: Add depth of core (and reason to stop drilling)**
Yes, see R1.13 above.

**P20, Fig. B2 caption. The peat in (A) would normally have a LOI of c. 95%. The default value of 80% does not generally apply**
Yes, see R1.9, above.

---

## Author Comment (AC2) · 29 Jun 2020

Reply to the interactive comment of Anonymous Referee 2 concerning the manuscript "Ground ice, organic carbon and soluble cations in tundra permafrost and active-layer soils near a Laurentide ice divide in the Slave Geological Province, N.W.T., Canada" by R.Subedi, S.V. Kokelj and S. Gruber.

We are grateful for the comments provided. Here, we respond to each issue raised and outline how the comments led us to revise the manuscript. The entire original text

of the interactive comment is shown in **bold font** and author responses in regular font. Each issue is identified with a code indicating "Referee 2" as well as a consecutive number, e.g., R2.1. This will help to revisit key issues raised by each of the three interactive comments in a summary reply that outlines the most important changes to the manuscript.

**0.1 Summary**

**This paper is formally well written and presented, and describes unique data from an interesting environment that is under-represented in the literature. I think that the data from these cores should be published, but in its current form the paper does not do the material justice. The paper suffers from a mismatch between its stated goals and its delivered conclusions. There are also problems with the methods that are major enough to call some of the conclusions into question. I recommend that the paper be rejected and returned to the authors for re-working and potential re-submission after the problems have been addressed.**

**0.2 Criticisms**

**0.2.1 Approach**

**The context for this study requires more thinking and description. For example, the authors state that, contrary to published maps, there is significant excess ice in the "area" – what is the area? The paper requires a map showing the Keewatin Ice Divide, the "central Slave Geological Province", etc. Table A.1 would be informative if it were a map or, even better, mapped together with an optical satellite image to allow comparison with land cover. For what region do the authors believe that their results are representative? Demonstrate that the set of**
**cores is somehow representative for a region and communicate what region this is. How much of the region that you so identify is represented by the four terrain types for which you present data?**

*R2.1 – Maps and context:* An overview map figure has been added to the manuscript and an additional map figure with the local study area to the new Supplement. We decided against optical satellite data as surface cover does not correlate well with sub-surface conditions.

**"Comparison with other permafrost regions..." is listed as the 3rd goal of the paper, but is not addressed and represents a central weakness. As an example, any reader of this paper would expect to find permafrost ONLY in those regions of the NWT and Yukon in which the authors have already worked, as well as in unglaciated portions of Alaska. Such myopia is not unique: many authors cite only papers within their national borders, but this criticism is a grave one for geoscientists, who should have a natural curiosity for the variability of expressions of their study objects on Earth. European, Alaskan and Russian research on permafrost is extensive, and offers many studies that deal with ALL of the study objects (glaciation, permafrost, ground ice, ground ice/water chemistry, etc.) and processes (thawing, landscape change, deglaciation, solute exclusion, thermokarst, etc.) in this study. Willfully ignoring most of the research in your field makes it impossible for the authors to demonstrate rigor in their approach, their knowledge of their field or an openness to the existing and potentially alternative interpretations of similar data sets. Comparing your results (e.g. Table D2) to only 2 previous studies (BOTH from your own research group!) cannot fulfill your stated goal to "...compare excess-ice content, organic-carbon density and soluble cation concentration with other permafrost environments...". It certainly reduces the relevance of your work to a broader readership and would alone be reason enough not to publish this manuscript.**

*R2.2 – Comparison with other permafrost regions:* The full text for Objective iii was "compare excess-ice content, organic-carbon density and soluble cation concentration with other permafrost environments *and with compilations such as overview maps and databases*". Here, in order to keep the scope of the manuscript manageable, "compilations such as overview maps and databases" have been taken as current reflections of what may be expected in the study area. To avoid misunderstanding, we have adjusted our formulation to read "... other permafrost environments OR with compilations ..." in the revised manuscript. Beyond that, we do find it unreasonable to conduct a study where conditions are contrasted between two regions, especially if one is well studied and is similar in some but not all aspects. This is part of formulating tractable questions.

*R2.3 – Comparison of solute contents with two studies is not enough:* Few other studies exist in formerly glaciated and nearby environments that can easily be compared. In part, this is due to the broad diversity of approaches in extracting soil water and reporting (normalizing) analytical results. This is also addressed in our responses R2.7 and R2.12, and now described in the revised manuscript. It also explains why two studies with overlap in authorship were chosen: here the methods are comparable. One study (Lacelle et al., 2019) has now additionally been quantitatively included as data was available digitally and could be converted. At the same time, the impact of differing lab methods is difficult to assess. Two additional studies from the vicinity of Yellowknife have also been included, although direct quantitative comparison is difficult.

**The study makes the a priori assumption that categorization of the soil cores by terrain types is justified and that averaging all borehole results within one terrain type is justifed. How were terrain types identified? Why were these four chosen? Why do the authors think that terrain type has an influence on ice content, organic carbon, and total soluble cations? Such questions are fundamental to study design and the means to reaching conclusions of broader**

**significance. Tellingly, there is no significance to the terrain types in the 6 con-clusions reached. At this point, the authors should have re-examined the basis for their choices. The results for all four parameters as a function of depth do not provide enough information (Fig. 3-6) for the reader to assess whether the cores really differ between terrain types. It is not clear why only total soluble cations are analyzed and presented. Why not other dissolved species? Why not present profiles for individual boreholes and/or cations? This work is left to the reader.**

*R2.4 – Justification of terrain types* The initial field sampling design and reasons for it having limited utility for interpretation of results is now described briefly in the revised manuscript. Based on this, the distinction of the four terrain types used is justified as a better way to group the locations analysed. Plots of individual borehole logs and analytical results are included in a new Supplement to prevent the manuscript from becoming too long. We do not agree that conclusions specific to each terrain type are a measure of their utility. Along the same lines, one could ask why individual boreholes should be shown. Terrain types are a useful grouping to add explanation to observations.

*R2.5 – Why only total soluble cations?* This study has unique data and insight to offer in several domains (ice, carbon, cations). As it is subject to a number of imperfections, focusing on total soluble cations and few salient features of their distributions keeps this robust and concise.

0.2.2   Methods & Data treatment

**The field photos of the cores are very impressive and show a wide variety of compositions and cryostructures – I think the paper would be strengthened by more of these and closer examination of the results. The methods used in this paper are beset with problems. The authors should refer to standard texts and**

**guides on soil analyses for methods of analyses. It is not sufficient to cite pre-vious studies by the same set of authors (using "cf."?) to establish that stan-dard accepted methods have been applied. Specific problems with the analytical methods are:**

*R2.6 – More core photos:* The published core photos are referenced in the description of the study region in the original manuscript. The data (more than 2.5 GB) is well organized by borehole and depth interval. Rather than include many more photos and increase the length of the manuscript, we point to this information with more speci-ficity in the revised version and point-out sections with telling photographs in borehole plots. Photos of frozen ice-poor till have now been included as an additional important example in a new figure.

*R2.7 – Standard accepted methods:* Janzen (1993) and Dean (1974), cited in the orig-inal manuscript, are standard texts. At the same time, these and other standard texts on soil analyses known to us do not describe methods for permafrost materials specif-ically. This is relevant because the extension of standard procedures (usually based in agricultural considerations) to permafrost materials is not straightforward. In many instances, no consensus on how this is best done is apparent from the permafrost lit-erature. Often, this problem is due to extreme ranges of water content when excess ice is present. For example, expressing water content on a dry gravimetric basis, as proposed in standard texts, can lead to high values that are difficult to interpret (Phillips et al., 2015). This is now also clarified in the revised manuscript.

**The are 2 sets of cation concentrations: those which underwent an arbitrary 1:1 dilution with water and those which did not. Diluting the solution affects the equilibrium between dissolved and adsorbed species. It is not acceptable to treat these data sets as equivalent expressions of some kind of concentration without at least testing for equivalence.**

[Figure]

*R2.8 – Comparing samples with and without added water:* In the revised manuscript, the known issues with differing extraction ratios are now described in the methods section. Additionally, Interpretation and Discussion now include test results about the effect of extraction ratio on our results.

**The creation of means as a function of depth is problematic and has not been justified – how is the 95% confidence limit calculated? In many cases MOST of the measurements lie outside of the 95% confidence limit. In what are the authors 95% confident? Anyone drilling in a similar location can be confident that most values lie outside of these limtis. With this kind of variability over depth, drawing a smoothed mean variation (the blue lines) over depth is nonsensical and masks real variability. It certainly requires examining each set of measurements on a per-borehole basis and on evaluating the data in detail.**

*R2.9 – Mean profiles:* Mean profiles make broad patterns emerging from multiple boreholes and/or samples more easily visible. Because the actual sample values are shown in the same graph, the real variability is not masked. The standard error of the average expresses the confidence in the average falling within this range, accounting for sample abundance and vertical extent. The majority of samples may be outside the standard error at 95% confidence for the mean profile.

**Similarly for Table D2: it is not clear how many samples are used to form the means, whether they are means, what the variability is, etc.**

*R2.10 – Data in Table D2:* The values from Kokelj and Burn (2005) are now identified as mean values in Table D2 of the revised manuscript. The number of samples or measures of spread are not included in the table of the original publication but, if desired, can be appreciated from the figures. Values from Kokelj et al. (2002) are now identified as 'estimated from figures'. For both, we prefer not to indicate statistical results that we estimate after the fact but rather report and interpret the publications' content in the

simplest way that supports our interpretation. The difficulty outlined here underscores the fact that finding more data for quantitative comparison is not straight forward.

**Presumably data from multiple cores is combined for each of the four regions – it is not clear and has not been established that these groups of cores are similar enough to be grouped, that the cores cover similar depth ranges, that the sampling frequency is similar, or that the sample sizes per terrain type have no effect. Present the data for individual cores, do not create means, etc. and then establish that the groups of cores are different using a test of significance. At the moment, all the work is left to the reader.**

*R2.11 – Combining multiple boreholes in a terrain type:* The justification for the terrain types is outlined in more detail in the revised manuscript and individual borehole plots are included in the Supplement. We also clarify that mean profiles are described for aiding quantitative description, rather than quantitative prediction, based on sometimes sparsely and unevenly sampled boreholes.

**The expression of concentration per dry weight of sediment is almost entirely meaningless. It is meaningless in terms of processes affecting concentration during freezing, thawing or in general in the field. Concentration gradients, moisture migration, and any other relevant processes will depend on concentrations in the pore space or pore water or liquid water. Concentrations should AT LEAST be reported in terms of the water volume obtained by thawing the samples.**

*R2.12 – Normalizing concentration per dry mass of sediment:* In the new Supplement to the revised manuscript, we now also plot concentrations in terms of the water volume obtained by thawing the samples, as requested (this was included in the supplementary materials digitally already before). Additionally, we explicitly mention the expression of results per dry weight in the conclusion to further prevent misunderstanding. We agree that this adds clarity as some results may depend on the method for normalization

chosen. We do not, however, agree with the broad assertion about the utility of the approach we have chosen: Solute contents in the permafrost literature are expressed in a variety of ways, for example with respect to dry mass, volume, or soil water content. None of these ways, similar to the choice of solute extraction (R2.7), is obviously perfect. In our example, standardization by dry weight makes sense because it helps in comparing permafrost (some very ice rich) and actively-layer soils (some coarse-grained and in dry convex upland locations). Furthermore, assuming that some of our ice rich sediments partially derive from Laurentide ice implies that the majority of solute found there can be assumed to originate from its particle content, possibly after thawing. For clarification, we have expanded the glaciological background so that the possibility of finding solute poor ice mixed with frozen sediments becomes more obvious. In summary, the issue of normalizing results, together with the wide differences in extraction methods used (see R2.7 – Comparing samples with and without added water) makes comparison of soil chemistry difficult between individual permafrost studies. To further clarify, a brief summary of these problems has been included in the manuscript.

**The calculation of excess ice content based on the ratio of volumes of thawed, saturated, settled sediment and supernatant liquid is problematic since the volume of supernatant liquid depends on soil texture.**

*R2.13 – Determination of excess-ice content:* Certainly. Further refinement or discussion of the shortcoming of this method, however, are beyond the scope of the work presented and would not affect our conclusions. The method we use remains the accepted standard in permafrost science (Subcommittee on Permafrost 1988) and engineering (ASTM 2016). Furthermore, line 124 in the original manuscript used 'estimate' to acknowledge that this is not clear cut.

[Figure]

**0.3 Conclusions**

**Each conclusion has problems:**

**1. Without placing your borehole sites in a geographical context, it is difficult to evaluate whether this qualifies as a new regional insight.**

*R2.14 – Geographic context and regional insight:* The context has now been expanded considerably with new figures and much expanded background on glaciological setting.

**2. The method of measuring excess ice does not allow the conclusion that thick occurrences of excess ice were found in tills. The photos show excess ice, but make it difficult to believe the volumetric values obtained by this method.**

*R2.15 – Excess ice amounts:* We maintain (see R2.12) that the method chosen for estimating excess ice content does allow the conclusion that thick occurrences of excess ice were found. We hope that the inclusion of individual borehole profiles in the Supplement and the published core photographs will also help alleviate this concern. We have addressed the question of the aerial abundance of thick excess ice now explicitly. While this is speculative, it helps to avoid the perception that we claim thick sequences of excess ice would be found everywhere. Finally, the amounts of 84% and 71% shown in Figure B3 were an error, as can also appreciated from Figure 4A that has no values above 80%. The values have now been corrected and additionally, the estimated visible ice content is shown per borehole, all of which are near 40%.

**3. The soil cores go down to ten metres and have maximal ice contents of 60%. If the deepest core had the highest ice content, you would have subsidence of less 6 m. How then is subsidence of "tens of metres" possible? Is this based on some kind of unmentioned extrapolation of observations?**
*R2.16 – Tens of meters:* The formulation of tenS of metres in line 326 was unintentional and has been corrected. Lines 9 (abstract) and 271, correctly read "metres to more than ten metres" and "up to more than ten meters".

**4. These are potentially interesting values, but would be made relevant if there was someway to know for what region the authors claim they are representative.**

*R2.17 – Area represented:* The field area shown on a map and specified by coordinates. The spatial aggregation of soil organic carbon storage has been dropped to keep the manuscript manageable.

**5. The cation concentration data are used only to establish in general "lower concentrations" when compared to a two studies from one other region and are entirely incidental to the paper's conclusions. There is no need to present terrain types, variation over depth or any of the data to reach this conclusion.**

*R2.18 – Conclusion concerning cation concentration:* Yes, we have chosen to remain with a simple indicator and analysis and a high level result that can be stated with some confidence. We report a summary that is true for all three terrain types that have mineral soils in permafrost and the active layer. We hope that the added detail in the explanation of terrain types will further alleviate the concern raised here.

**6. I agree that geological legacy is important. The data here are insufficiently linked to geological legacy.**

*R2.19 – Linking with geological legacy:* This should be more obvious now with more explicit geographic context, justification of terrain types, and individual profile data shown and discussed.

**1 References**

Phillips, M R, C R Burn, S A Wolfe, P D Morse, Adrian J. Gaanderse, H. B. O'Neill, D.H. Shugar, and S. Gruber. 2015. "Improving Water Content Description of Ice-Rich Permafrost Soils." In Proceedings of the GeoQuebec 2015 Conference, September 20-23, Quebec, Canada.

Subcommittee on Permafrost 1988: Glossary of permafrost and related ground-ice terms. Associate Committee on Geotechnical Research, National Research Council of Canada, Ottawa.

ASTM 2016: D4083-89(2016) Standard Practice for Description of Frozen Soils (Visual-Manual Procedure). West Conshohocken, PA; ASTM International, 2016.

---

## Author Comment (AC3) · 29 Jun 2020

Reply to the interactive comment of Dr. S. Wolfe concerning the manuscript "Ground ice, organic carbon and soluble cations in tundra permafrost and active-layer soils near a Laurentide ice divide in the Slave Geological Province, N.W.T., Canada" by R.Subedi, S.V. Kokelj and S. Gruber.

We are grateful to Dr. Wolfe for his comments. In this reply, we outline how the comments have led us to make corrections, clarifications or additions to the manuscript and

also, we provide specific explanation where we disagree with the statements made in the comment. The entire original text by Dr. Wolfe is shown in **bold font** and author responses in regular font.

**1 Introduction**

**The paper under review by Subedi, Kokelj and Gruber provides data on ground ice, organic carbon, and soluble cations from drill holes in the Lac de Gras region of the Slave Geological Province, Canada. The authors indicate that their results differ from observations made in western Arctic Canada and they make specific comparisons to field studies within the Mackenzie Delta region. They further indicate that the study provides quantitative data for a region that has few previous studies.**

**However, several statements and conclusions by the authors require reappraisal and revision in light of existing literature. Three issues in particular merit attention. Regarding the glacial context, the authors contend that the site is near the Keewatin Ice Divide, although it is more than 500 km distant. In reporting ground ice, they interpret model outputs prepared at a national scale in a local context, and combine surficial units with critically different properties. Lastly, the authors overlook relevant studies published from the Slave Geological Province, including from the study area.**

**Literature from the Slave Geological Province is discussed here to assist the authors in their task.**

**2   Glacial Context**

**The authors represent the study site as near a Laurentide Ice divide and having been influenced directly by it when, in fact, the Keewatin Ice Divide is distal to the study area.**

**The title of the paper indicates that the study area is "near a Laurentide ice divide", while line 29 states that the Lac de Gras region "is situated close to the Keewatin Ice divide of the Laurentide Ice Sheet". The term "Laurentide ice divide" should not be used. "Laurentide Ice sheet ice divide" or" Keewatin Sector ice divide" are appropriate alternatives.**

**Laurentide Ice Sheet ice divides comprise named ice divides (with capital letters: Keewatin Ice Divide, Labrador Ice Divide, M'Clintock Ice Divide, etc.) and unnamed ice divides (see Dyke and Prest, 1987). However, the Keewatin Ice Divide (KID) is by definition "the zone occupied by the last glacial remnants of the Laurentide Ice Sheet west of Hudson Bay" (Lee et al., 1957). Therefore, the KID was situated at least 500 km east of the study area (e.g. McMartin and Henderson, 2004). All the other ice divide positions in Keewatin are not located closer to the study area and cannot be termed "Keewatin Ice Divide".**

*W1 – Is the study area near a Laurentide ice divide?* We take a broad view of the literature in geology and glaciology, and of the temporal extent during which the Laurentide Ice Sheet (and possible earlier continental ice sheets) has shaped the landscape we observe today. Ice divides move over time (Benn & Evans 2010) and angles between preserved flow features are large in the zone of shifting ice divides (Boulton & Clark 1990a). This is consistent with the differing directions of ice flow (southwest, west, and northwest) field-mapped in the study area (Dredge et al., 1994). Furthermore, simulation (Margold et al., 2018) and mapping (Boulton & Clark, 1990a) results suggest evolving positions of ice divides, with some closer than 500 km to the study area. Predominant zones of erosion and deposition are apparent at the continental scale

and have been hypothesized to be related with basal thermal zones and ice divides during glacials (Sugden 1978; Benn & Evans 2010; Boulton & Clark 1990b). These zones emerge in the long term even though ice divides move laterally in response to the evolution of ice sheets. Other work on surficial geology takes a similarly broad view and includes our study sites, for example Figure 3 "Generalized bedrock geology of the area around the Keewatin Ice Divide" of Aylsworth and Shilts (1989). Being 'near' an ice divide expresses the quality of neither being near the margin of the ice sheet, nor beneath the ice divide, but rather in an inner area influenced by proximity to a spreading centre and increasing net erosion. The proximity to a spreading centre is not expressed well by using 'central' or 'middle' as these terms bear no relationship to overall ice flow and are misleading in an ice sheet with multiple domes. Nevertheless, the term 'near' is ambiguous and we agree that this is difficult. A new figure and more explanation on the glaciological background relevant to clarify this will be added in the revised manuscript. Based on this context, the actual distance to the approximate ice divide / spreading centre will be stated instead of 'near' throughout most of the revised manuscript.

*W2 – The term "Keewatin ice divide"* We will omit the term "Keewatin Ice divide" (partially capitalized by mistake) to avoid an interpretation in terms of the field-mapped final position defined by (Lee et al., 1957). Instead, we will reformulate these sentences with reference to the "Keewatin Dome spreading centre". For the title, however, we prefer to keep the term "ice divide" because it is clearer in concisely expressing a relation to overall ice flow than the concepts of ice domes or spreading centres would be.

*W3 – "Laurentide Ice sheet divide" vs "Laurentide ice divide":* The comment does not state why "Laurentide ice divide" should not be used and "Laurentide Ice sheet ice divide" or" Keewatin Sector ice divide" are appropriate alternatives. Misinterpretation with respect to present day ice divides on the Canadian Shield is unlikely based on the specifier "Slave Geological Province, N.W.T., Canada", a region where no glaciers

currently exist. We prefer to keep the concise title.

**If the authors wish to consider features 500 km from their study area as proximal, then a reader would further expect literature that reports conditions in the glacial sediments near Yellowknife, only 310 km from the study area (line 79), to be fully considered. This is addressed further below.**

*W4 – Inclusion of environments near Yellowknife:* The nature of the deposits and permafrost history near Yellowknife differ strongly from those in our study area. Comparing ('fully considered') across a transect in temperature conditions in addition to proximal versus distal location with respect to the Keewatin Dome spreading centre would complicate the manuscript. We now include two additional references from the Yellowknife area in the revised manuscript to provide context.

**Basal ice sheet conditions, as discussed by Rampton (2000) and Utting et al. (2004), influence the source materials for shield-derived tills of the Slave Geological Province. Glacial conditions in the Slave Geological Province differed significantly from those of western Arctic Canada. Such differences had a profound effect on ground ice development (Wolfe et al., 2017). The authors compare their results with conditions in western Arctic Canada, where the source materials for till were derived from the sedimentary basins of the Interior Plains as opposed to the Canadian Shield. It is not evident why this setting is the primary reference for comparison with results from this study without relevant details on conditions in the Slave Province.**

*W5 – Why is the study area compared with conditions in western Arctic Canada?* Western Arctic Canada is one of the most intensively studied areas and, as such, a natural point of comparison, especially with respect to solute concentrations in permafrost and the active layer. This is mentioned multiple times in the manuscript. In the revised version, we have added more detail on basal ice sheet conditions and the differences

to be expected between differing zones. Additionally, we now reference Gaanderse et al. (2018) for context with shield-derived sediments in the Slave Geological Province (see W14, below).

**3  Ground ice reporting**

**3.1  Use of model outputs**

**The authors apply national-scale modelling results to local site conditions.**
*W6 – national-to-local comparison:* This is not what we do. Local application would suggest extracting model values for the locations of boreholes. By contrast, we have been careful to avoid this, and by using a 50 km x 50 km area, we express the fact that what is found in the ground is not reflected in about 2500 model cells of 1 km x 1 km in the vicinity. The corresponding discussion has been expanded and a table added to prevent this misunderstanding.

**The authors state that "the new Ground Ice Maps for Canada (O'Neill et al., 2019) show the study area (50 km x 50 km) to contain no or negligible wedge ice, negligible to low segregated ice and no relict ice, which includes buried glacier ice." (lines 40- 42). In fact, the new Ground Ice Maps for Canada depict wedge ice from none to low; segregated ice from none to medium, and relict ice (which includes buried glacier ice) from none to low (Figure 1). The authors thereby under-report the amount of ground ice depicted for the study areas by O'Neill et al. (2019).**
*W7 – Under-reported ground ice depicted by O'Neill et al. (2019):* The values we report are true to those shown in O'Neill et al. (2019). The points shown by Figure 1 of Dr. Wolfe's comment do not match the locations we report in Table A1 of the original manuscript. We have added a map and more details to the revised manuscript to

underpin the values we report.

**Nevertheless, differences between the authors' reporting and the model results are due, in part, to site-specific surficial geology. The surficial geology used in modelling is at scale of 1:5 M (GSC - Surficial Geology of Canada, 2014). For associations between ground ice and surficial geology to be appropriately considered at the local scale of 1:125,000, Dredge et al. (1995) and Ward et al. (1997) may be consulted.**

*W8 – Appropriate scale of application for the map by O'Neill et al. (2019):* We fully agree with this statement. However, this scaling problem is best accounted for at the stage of model building, not in comparison with evidence. Coarse-scale maps must provide a useful generalization of ground conditions to have value for practical purposes and as scientific tools. If comparison with observations were deemed inappropriate, we would be on the slippery slope to an irrefutable hypothesis. The comparison of a map or model with observations is one of their key advantages. We agree that it must happen at, and appropriate to, the level of generalization used. In our manuscript, we compare the overall picture given by the map and not local patterns or point locations. In our 50 km x 50 km area, all 2500 1-km cells have class 'none' (not 'negligible') for ground ice. The entire Lac de Gras map sheet 76-D by Ward et al. (1997) (cited in the manuscript as "Geological Survey of Canada, 2014" in an updated version) has 97% of its cells classified as 'none', 3% cells classified 'low' and 0.08% classified as 'medium'. The adjacent Aylmer Lake map sheet 76-C by Dredge et al. (1995) is to 99.9 % classified as 'none'. Application of the mapping rules in O'Neill et al. (2019) at the scale of 1:125,000 as suggested would likely lead to drastically different abundance for relict ice. As such, our results may inform improved ground-ice mapping simply by bringing attention to this. In the revised manuscript, we now discuss this scaling issue and how it can be addressed in model applications based on the nationwide surficial geology at the scale of 1:5 M.

**3.2 Use of surficial geological units**

**The authors combine surficial units with critically different properties in the context of reporting excess ice.**

**The authors combine drill cores into "upland tills", which they define as "smoothly rounded hills comprised of thick till and in till veneer over bedrock" (line 187-188). Combining drill cores from till veneers, which are tills that are less than 2 m in thickness, with the drill cores of thick till misrepresents the extent of "upland till" and therefore of ground ice contained within till terrain. To this end, Ward et al. (1997) and Dredge et al. (1995) provide more suitable spatial depiction of till veneer, till blankets, and hummocky till that permit the drill cores to be allocated to these specific till units. Such separation is appropriate for depicting depth versus water content and excess ice (as in Figure 3A and 4A). This approach may highlight the lower excess ice abundance in till veneers and at depths above 4 m, and higher amounts in thicker tills (and at depths below 5 m). These data may further inform understanding of the proportion of hummocky till, and thus potentially preserved Laurentide basal ice, in the area.**

*W9 – Misrepresenting the extent of upland till and therefore ground ice:* We did not perform spatial aggregation of ground ice contents, nor did we state a spatial extent for upland tills. In the revision, we add clarity by always referring to our terrain type as 'upland tills' to avoid the ambiguity that existed when using the contracted 'tills'. Additionally, we now provide a first-order estimates for the spatial extent of upland tills with relict ice.

*W10 – Combining the mapping units of 'till veneer', 'till blanket' and 'hummocky till' into 'upland tills':* The mapping units of 'till veneer', 'till blanket' and 'hummocky till' are interpretations of till thickness and spatial extent derived largely from its surface expressions on aerial photographs. The lines drawn thereby represent a decision made

by an experienced mapper, consistent with the level of spatial generalization required for a particular scale but not the actual conditions in the field. Our original field strategy followed these mapping classes and the map by Ward et al. (1997), as here proposed by Dr. Wolfe. The results however have shown that there is a difference between theory (or mapping) and reality (or field) and this has led us to use terrain types more suitable to conceptualize and interpret out results in the manuscript. In the revised manuscript, we now explain the original sampling approach briefly, as well as the reasons for deviating from the map units shown in Ward et al. (1997).

**In addition, in presenting "fluvially reworked till (the Valley)" (line 185) where "silts and sands are well sorted and likely derived from fluvial reworking of local tills" (line 191) and in presenting evidence of post-glacial ground ice melt features (e.g. Figure C1) the authors might acknowledge alternative interpretations by Rampton (1999) and by Utting et al. (2009) to account for fluvial activity. Alternative classification of the terrain types is required because the current terms conflate categories of phenomena, e.g. till and valley.**

*W11 – Alternative interpretations:* The interpretation of Rampton (2000) has been included, assuming that "Rampton (1999)" is a typo and in fact, refers to Rampton (2000). Utting et al. (2009) study glaciofluvial corridor hummocks; the relation to this manuscript is unclear.

*W12 – Conflation of phenomena:* The original manuscript consistently uses "the Valley", with both the definite article ("the") and capitalization indicating that this refer to one specific valley, as described in Section 4.1. For increased clarity, we now discuss in more detail how this terrain type differs from other locations.

**4  Incorporation of comparative literature**

**The authors overlook existing regional and local literature with similar conclusions, thereby claiming undue precedence.**

**4.1  Solutes in mineral soils**

**The authors' state that "the concentration of total soluble cations in mineral soils is much lower than at other previously studied locations in the western Canadian Arctic" (lines 14-15) and that "the absolute concentrations of soluble cations obtained in the study area near Lac de Gras are low compared to previous studies from northwestern Canada that report higher concentrations in active layer and permafrost across diverse terrain types (Table D2)." (lines 302-304). These remarks assume that all comparable previous studies have taken place in the Mackenzie delta area or Herschel Island (Table D2). The authors indicate that "The low concentrations in our study area are associated with the contrasting nature and origins of surficial materials. Tills in our study region are generally coarser grained than many glacial deposits studied in the western Arctic, are regionally sourced from mostly granitic rocks and have been exposed only to minor postglacial landscape modification (Haiblen et al., 2018; Rampton and Sharpe, 2014)". (lines 311-314).**

*W13 – Assumption that all comparable previous studies have taken place in the Mackenzie delta area or Herschel Island:* We disagree with this assertion. The statements that Dr. Wolfe cites are true based on the two (now three) studies we compare with. It is not the remarks that do the assuming of 'all'.

**Gaanderse et al. (2018) originally reported on solute concentrations from glaciolacustrine deposits within the shield area that indicate low values similar to the**

**Lac de Gras area. Gaanderse et al. (2018, p. 1039 noted that "Total soluble ions concentrations decreased with depth from the active layer to the underlying glaciolacustrine clays in permafrost (Figure 8). This trend contrasts with observations in the western Arctic, where low ion concentrations occur within sediments of the active layer and near-surface permafrost, relative to the underlying permafrost (Kokelj et al., 2002; Kokelj and Burn 2003, 2005; Lacelle et al., 2014). Unlike the predominantly marine origin and the mixed-layered clays of the western Arctic (Dewis et al., 1972; Kodama, 1979), the glaciolacustrine clays of the Great Slave region are not inherently solute-rich or weathered." And "These fine-grained glaciolacustrine, lacustrine and alluvial sediments belong to the same generation of glacially-derived sediments with a regional mineralogical composition from igneous and metamorphic sources (Aden et al., 2015). The clays and clay-sized glaciolacustrine sediments are predominantly unweathered, with major soil ion abundances likely reflecting the mineralogy of local rocks, including Ca2+, Na2+, and K+ from the weathering of feldspars; Mg2+ and Ca2+ from amphiboles, pyroxenes and olivine; S042- from sulphides and Cl- from igneous sources."**

*W14 – Gaanderse et al. (2018)* We now reference Gaanderse et al. (2018) to provide context of low solute concentrations in shield-derived sediments from the Slave Geological Province. Two caveats remain: (1) the nature of the deposits and permafrost history differ from the Lac de Gras area and (2) quantitative comparison is difficult because they extract and report solutes with respect to saturated paste and we do not know the saturated-paste water content for our samples. We now give more details on the various methods of extracting and reporting solute contents to further explain why few studies are used for comparison and to provide context for these caveats.

**Additional supporting data on low concentrations of soluble cations from the Slave Geological Province and in the Lac de Gras area are also presented in Wolfe et al. (1997a) and Wolfe (1998) who describe low cation concentrations**
**in buried ground ice in glaciofluvial delta sediments, and in Wolfe et al. (1997b), who include borehole logs, geophysical surveys, cation concentrations and oxygen isotopes from a 14-m borehole at the BHP Airstrip Esker within the authors' Lac de Gras study area.**

*W15 – additional data published by Dr. S. Wolfe:* In the revised manuscript, we have now included a reference to the solute concentrations summarized in Wolfe (1998), although the comparison requires some assumptions about the method of water extraction and reporting. Wolfe et al. (1997a) describe oxygen isotopes and cation concentrations for ground ice, whereas our contribution describes soils samples. It is difficult to make a meaningful comparison with this contributions. Its insight that massive ice exists in glaciofluvial sediments in the area is already included in our manuscript via the citation to Dredge et al. (1999) who summarise key insights on ground ice from this work, as well as from BHP Diamonds Inc (1995) and EBA (1995). Wolfe et al. (1997b) appears to be a work report that was then distilled in the other two papers by its authors.

4.2   Ground ice expectations

**The authors state that "permafrost in the study area contains much more ground ice than expected" (line 16). As noted above, several studies have illustrated the presence of high ground ice contents in outwash sediments in the area. In addition to these, Dredge et al. (1999), referenced by the authors, clearly present expected ground ice conditions and terrain sensitivity in line with the authors observations. The importance of the geological legacy in determining the characteristics of permafrost and potential responses of this system of disturbance and change is further summarized in Wolfe et al. (2017), who conclude that "Glacially-derived ground ice includes buried glacial ice within glaciofluvial outwash deposits and buried glacial and meltwater ice**

**within eskers. Sediment-rich ice has also been encountered within hummocky till terrain during mine development operations. Surficial features attributed to partial thawing and creep of massive ground ice are regionally apparent. Although massive ice has been encountered in only a few locations to date, buried ground ice may be common within this glaciated region of the Tundra Shield."**

*W16 – Is there more relict ground ice in the study area than expected?* The new ground ice maps for Canada by O'Neill et al. (2019) show the project area, and most of its surroundings (see comment W6), as having no relict ice. This is despite the alternative classes 'negligible' and 'low' being available in the map. The rules formulated in a model and the results accepted in its evaluation express the expectations held by the authors (including Dr. Wolfe) and accepted by its internal and external reviewers. Similarly, a user of this map that is available digitally, will form their expectations based on what the legend states. These expectations match with those formulated in earlier maps but not with our findings.

**Nevertheless, the authors are still cautioned about asserting that "Tills in our study areas . . . have been exposed to only minor post-glacial modification (Haiblen et al., 2018; Rampton and Sharpe, 2014)", noting the evidence of Holocene warming and tree-line advance in the region as noted by the authors (lines 83-84) and in Moser and MacDonald (1990) and MacDonald et al. (1993).**

*W17 – Post-glacial modification:* Yes, this sentence applies to the majority of only UPLAND tills and has been clarified accordingly.

**The authors state in the abstract that "thaw subsidence of metres to more than ten metres is possible" due to ground ice that may be buried Laurentide**

basal ice (line 8 – 9). **Within the paper, the authors write: "A potential surface lowering of many metres, up to more than ten metres, is thus to be expected from areas of thick till if this permafrost was to thaw completely" (lines 271-272). Again, in the conclusions, the authors state: "Thaw-induced terrain subsidence on the order of metres to tens of metres is possible in ice-rich till" (line 326). The statements are based upon data from only one borehole with samples from below 6 m depth. The borehole terminated at 9.5 m depth. The authors assume that conditions in this borehole are representative of all till "estimated to be 10-30 m thick in the area (Haiblen et al. 2018)" (line 270). In other words, the authors assume, without disclosed evidence, that the excess ice profile presented in Fig. 4 extends indefinitely downwards with high values, that it applies consistently throughout an extensive till unit, and that the unit is sufficiently thick to contain excess ice tens of metres thick. Readers should be made aware of the assumptions upon which these statements are based and, in particular, should be able to recognize that the principal data contributing to these assertions are derived from 3.5 m of drill core, from 6 to 9.5 m in the profile.**

*W18 – TenS of meters:* The formulation of tenS of metres in line 326 was unintentional and has been corrected. Lines 9 (abstract) and 271, correctly read "metres to more than ten metres" and "up to more than ten meters".

*W19 – Only one borehole with samples from below 6m depth:* More detail supporting our conclusion has been inserted in the revised manuscript: (1) Table A1 now shows that 5/10 boreholes in upland tills terminate in bedrock or a boulder and the remaining 5/10 terminate in ice rich material. This supports the expectation of some continuation with depth. (2) Information for individual boreholes in now included as a plot in Supplementary Materials. (3) One more borehole from an earlier campaign has been added to the discussion. The geomorphological argument in Figure C1, although hypothetical, adds some further support.

*W20 – Assumptions of the authors:* We do not hold the assumptions ('representative of all till', 'extends indefinitely downwards', 'applies consistently throughout an extensive till unit') ascribed to us here. In the revised manuscript we improve clarity by including an estimate of plausible ranges for the extent of relict ice. While uncertain, this will prevent misunderstanding.

**5 Summary and Conclusion**

**The paper by Subedi, Kokelj and Gruber (in review) provides an added contribution to the growing knowledge of permafrost and ground ice in the Slave Geological Province. The purpose of these comments is to provide an appropriate regional context for the observations. The authors may take advantage of these comments so that their contribution to the literature will complement, and be informed by, the existing knowledge of permafrost, environmental change and ground ice conditions in this region.**

**6 Acknowledgements**

**These comments benefitted from discussion and input with several colleagues. In particular, Drs. Chris Burn, Brendan O'Neill, Peter Morse, Dan Kerr and Isabelle McMartin are gratefully acknowledged.**

**7  References**

Aden, A.A., Wolfe, S.A., Percival, J.B. and Grenier, A.: Characteristics of glacial Lake McConnell clay, Great Slave Lowland, Northwest Territories; Geological Survey of Canada, Current Research 2015-7, 12 p, 2015.

Dredge, L.A., Ward, B.C. and Kerr, D.E.: Geological Survey of Canada, "A" Series Map 1867A, 2 sheets https://doi.org./10.4095/207631, 1995.

Dyke, A. and Prest, V.: Late Wisconsinan and Holocene history of the Laurentide ice sheet. Géographie physique et Quaternaire, 41, 237-263, 1987.

Gaanderse, A.J.R., Wolfe, S.A., and Burn, C.R.: Composition and origin of a lithalsa related to lakeâARlevel recession and Holocene terrestrial emergence, Northwest Territories, Canada. Earth Surface Processes and Landforms 43, 1032-1043, 2018.

Geological Survey of Canada.: Surficial Geology of Canada. Geological Survey of Canada, Canadian Geoscience Map 195, (ed. Prelim., Surficial Data Model V.2.0 Conversion), 1 sheet, https://doi.org/10.4095/295462, 2014.

Lee, H.A., Craig, B.G. and Fyles, J.G.: Keewatin Ice Divide. Geological Society of America, Bulletin 68, 1760-1761, 1957.

MacDonald, G.M., Edwards, T.W.D., Moser, K.A., Pientiz, R., and Smol, J.P.: Rapid response of treeline vegetation and lakes to past climate warming. Nature, 361, 243-246, 1993.

McMartin, I. and Henderson, P.J.: Evidence from Keewatin (central Nunavut) for paleo-ice divide migration. Géographie physique et Quaternaire, 58, 163-186, 2004.

Moser. K.A. and MacDonald, G.M. : Holocene vegetation change at treeline north of Yellowknife, Northwest Territories, Canada. Quaternary Research, 34, 227-239, 1990.

O'Neill, H.B., Wolfe, S.A. and Duchesne, C.: New ground ice maps for Canada using a

paleogeographic modelling approach. Cryosphere 13, 753–773, 2019.

Rampton, V.N.: Large-scale effects of subglacial meltwater flow in the southern Slave Province, Northwest Territories, Canada Can. J. Earth Sci. 37, 81–93, 2000.

Utting, D.J., Ward, B.C. and Little, E.C.: Genesis of hummocks in glaciofluvial corridors near the Keewatin Ice Divide, Canada. Boreas, 38, 471-481, 2009.

Ward, B.C., Dedge, L.A. and Kerr, D.E.: Geological Survey of Canada, "A" Series Map 1870A, 2 sheets, https://doi.org/10.4095/209260, 1997.

Wolfe, S.A.: Massive ice associated with glaciolacustrine delta sediment, Slave Geological Province, N.W.T., Canada. In: Proceedings Seventh International Conference on Permafrost, Yellowknife, NWT. A.G. Lewkowicz and M. Allard (eds). Collection Nordicana, University Laval, 1998.

Wolfe, S.A., Kerr, D.E. and Morse, P.D. 2017.: Slave Geological Province: an archetype of glaciated shield terrain. In: Landscapes and landforms of western Canada. O. Slaymaker (Ed). World Geomorphological Landscapes. Springer International Publishing, Switzerland.

Wolfe, S.A., Burgess, M., Douma, M., Hyde, C., Robinson, S.: Geological and geophysical investigations of ground ice in glaciofluvial deposits, Slave Province, District of Mackenzie, Northwest Territories.; in Current Research 1997-C: Geological Survey of Canada, 39-50. 1997a.

Wolfe, S.A., Burgess, M.M., Douma, M., Hyde, C. and Robinson, S.: Geological and geophysical investigations of massive ground ice in glaciofluvial deposits, Slave Geological Province, Northwest Territories. Geological Survey of Canada, Open File 3442. 50 p. 1997b.

Additional references for consideration: BHP Diamonds Inc.: NWT Diamonds Project environmental impact statement. Vol. 1-4. BHP Diamonds Inc., Vancouver, B.C., 1995.

EBA Engineering Consultants Limited: Koala mine airport esker evaluation. EBA Report No 0101-94-11439.3, submitted to BHP Diamonds Incorporated, Vancouver, B.C., 1995.

**8 New references**

Aylsworth, J.M., and W.W. Shilts. 1989. "Glacial Features around the Keewatin Ice Divide: Districs of Mackenzie and Keewatin." Geological Survey of Canada, no. Paper 88-24: 21 p.

Benn, D., Evans, D. J. (2014). Glaciers and glaciation. Routledge.

Boulton, G. S., and C. D. Clark. 1990a. "A Highly Mobile Laurentide Ice Sheet Revealed by Satellite Images of Glacial Lineations." Nature 346 (6287): 813–17. https://doi.org/10.1038/346813a0.

Boulton, G. S., and C. D. Clark. 1990b. "The Laurentide Ice Sheet through the Last Glacial Cycle: The Topology of Drift Lineations as a Key to the Dynamic Behaviour of Former Ice Sheets." Transactions of the Royal Society of Edinburgh: Earth Sciences 81 (4): 327–47. https://doi.org/10.1017/S0263593300020836.

Dredge, L.A., B.C. Ward, and D.E. Kerr. 1994. "Glacial Geology and Implications for Drift Prospecting in the Lac de Gras, Winter Lake, and Aylmer Lake Map Areas, Central Slave Province." In Current Research 1994-C; Geological Survey of Canada., 33–38.

Margold, Martin, Chris R. Stokes, and Chris D. Clark. 2018. "Reconciling Records of Ice Streaming and Ice Margin Retreat to Produce a Palaeogeographic Reconstruction of the Deglaciation of the Laurentide Ice Sheet." Quaternary Science Reviews 189: 1–30. https://doi.org/10.1016/j.quascirev.2018.03.013.

Sugden, D. E. 1978. "Glacial Erosion by the Laurentide Ice Sheet." Journal of Glaciol-

ogy 20 (83): 367–91.

---

## Author Comment (AC4) · 29 Jun 2020

This is a high-level summary of the changes made in the revised manuscript, important relations with comments by Referees 1 and 2 and by Dr. Wolfe are referenced (R1, R2, W). We have expanded several sections with additional detail and the manuscript has increased in length and complexity. We undertook to preserve simplicity by creating a detailed Supplement and by minor reorganization and editing.

[Figure]

**1 Regional context and characteristics of study area**

Regional context has been expanded and illustrated with a new figure (R1.2, R1.4, R2.1, R2.14, R2.19, W4). The abundance of surficial geology classes is now tabulated for the study area and two surrounding map sheets (R1.14).

**2 Glaciological context**

More detailed explanation of the glaciological context and a new figure have been added. With this, the ambiguous notion of being 'near' an ice divide is now specified further in terms of distance and put into a geomorphic and glacial context (W1).

**3 Information for individual boreholes**

We have expanded Table 1 and added a Supplement that provides a detailed map of the study area, an overview plot comparing all boreholes, and individual plots of major results for each borehole (R1.5, R1.13, R1.15, R2.14, W19). The publicly available core photos are mentioned explicitly and one new figure with core photos has been added (R2.6). The original (but abandoned) sampling strategy has been described and the four terrain types we now use instead are justified (R2.4, R2.11, W10).

**4 Accounting for large clasts in aggregated organic-carbon densities**

We have derived the approximate proportion of large clasts from the core photographs and applied this to the aggregation of organic-carbon densities. The has lowered the

values we report by up to 7%. (R1.8)

**5   Comparison of cation concentrations with other studies**

More background on the difficulties of comparing studies in the absence of consensus methods is now outlined (R2.7, R2.8, R2.12) and more studies, also from environments with similar material origins but differing depositional history, have been added (R2.3, W15).

**6   Spatial abundance and mapping of preserved relict ground-ice**

We now provide spatial aggregates of relict ice predictions by a previous model in a new table (W6, W7). We also estimate plausible ranges for the the spatial extent of relict ice (W20). We discuss the difference between our findings and a published prediction of relict ice abundance more explicitly and explain how the underlying scaling issue can be addressed to improve future models (W8, W16). The importance of the 'mosaic' character of the landscape is now emphasised more clearly throughout the manuscript, accordingly.

**7   Conclusions**

Our conclusions remain largely unchanged. The statement on spatially aggregated soil-organic carbon storage has been omitted, along with its background in the manuscript, to keep the manuscript length and complexity manageable.

---

## Author Response (AR2)

**Reply to review comments**

Stephan Gruber

October 5, 2020

Reply to Anonymous Referee 1 concerning the revised manuscript with the final title "Ground ice, organic carbon and soluble cations in tundra permafrost soils and sediments near a Laurentide ice divide in the Slave Lake Geological Province, N.W.T., Canada" by R. Subedi, S.V. Kokelj and S. Gruber.

We are grateful for the Referee's second round of constructive and detailed comments. Here, we respond to each issue raised and outline how the comments led us to revise the manuscript. The entire original text of the interactive comment is shown in **bold font** and author responses in regular font.

A version of the manuscript with highlighted changes, red for deletions and blue for insertions, is provided for convenience. During the revisions, we have made a number of small additional improvements to spelling and clarity. In the highlighted manuscript, these changes are indicated, magenta for deletions and cyan for insertions.

**0.1 General comments**

**In my original review of this manuscript I suggested minor revisions. On the one hand, this study presents unique and highly valuable data on ground ice, organic carbon and soluble cation contents in permafrost cores of the Central Canadian Arctic. On the other hand, I had questions related to representativity of the study area for the wider region as well as the thoroughness of field and laboratory procedures. In my opinion, the overall value of the dataset outweighs uncertainties associated with some methodological shortcomings, which to a large extent cannot be corrected any longer.**

**In the revised version of the manuscript, the authors have addressed the representativity of the study area by comparing surface geology maps for the study area and two larger sheets. Unfortunately, the four landscape classes identified in this study cannot be directly linked to the units in the surface geology maps, and their proportional extent is unclear.**

**As stated above, many of the methodological issues cannot be remedied at this late stage, but the authors make generally a bet-**

ter case of identifying uncertainties derived from these shortcomings (e.g., lack of field volume for collected samples, a single LOI value for organic samples, volume occupied by large clasts, the role of the 0.5-5 mm fraction, etc.).

The added Supplement with individual core logs is highly valuable.

**0.2   Specific comments**

**Title: The title has become too long and is partly inaccurate. It is unnecessary to add ..., including preserved Laurentide basal ice, ... The polygenetic nature of the ground ice is clearly explained in the abstract as well as in the results. There is nothing like ...   active-layer soils ...   It is implicit that a permafrost soil has an active layer. I suggest, therefore to change the title to 'Ground ice, organic carbon and soluble cations in tundra permafrost soils and sediments near a Laurentide ice divide in the Slave Lake Geological Province, N.W.T., Canada'**
Changed as suggested.

**Page 1, Line 13 (P1, L13): ..., based on greater spatial and ...**
Changed as suggested.

**P1, L16: ...   lower than at previously studied ...**
Changed as suggested.

**PP1, L18: No need to start a new paragraph**
Changed as suggested.

**P1, L19: The last sentence is out of scope.  This study does not present any evidence for buried paleosols of interglacial/interstadial age, which are not to be expected due to the (at times) warm-based erosive nature of the Ice Sheet in this region**
Omitted "as well as environmental conditions during the last interglacial."

**P2, L23: ...  It sampled the active layer and permafrost layer of soils and sediments and ...**
Changed as suggested.

**P2, L33+: ...   The area beneath the Keewatin Dome spreading centre (Fig. 1C, zone 1), ...**
Changed as suggested.

**P2, L36: ...   in the western Canadian Arctic (Fig. 1C, zone 3). Between both (Fig. 1C, zone 2) is an area of increasing ...**
Changed as suggested.

**P3, L62: ecological succession = peat accumulation**
Changed as suggested.

**P5, Fig. 1: Explain zones 2 and 3 in caption figure 1C** Clarification and explanation has been inserted.

**P5, L123: Soil parent materials consist of ...**
Changed as suggested.

**P9, L196: 'homogenized'. Here it should be explained that dried samples were crushed, and that any larger organic fractions (>0.5mm) like roots, plant remains are therefore part of the P0.5 sample analyzed for LOI (see answer of authors to my original question)**
Inserted "Because homogenization involved crushing the oven-dried sample with mortar and pestle, any larger organic fractions like roots and plant remains are therefore part of the sample analyzed for LOI.".

**P9, L202: ... This occurred only in the top 0.3m ? (but also in peat down to > 2m ?)**
We confirm that the LOI estimation of 80% has been applied only in the top 0.3m and there, with one exception (sample NGO-DD15-1010-01 at 0–0.15m), only in pit samples.

**P13, L266: It is not at all evident from figures 5 and 6 that cation values are lower in organic/active layer vs mineral/permafrost layer. Please check this sentence/these numbers**
The sentence is correct as is. It expresses the relationship at the level of samples (not terrain types) and the distinction between organic and mineral is described in the preceding sentence.

**P15, L283: ... Figure B2 panels B) and C) ... Incomplete sentence**
We have clarified this sentence.

**P16, L288: growth = accumulation**
Changed as suggested.

**P17, L332: Using samples from the Greenland ...**
Changed as suggested.

**P19, L393: Table D1 compares the SOC storage in till soils (considered Turbels), esker soils (considered Orthels) and organic soils (considered Histels) to published values (Hugelius et al., 2014). This is useful but it should be considered that these are circumpolar mean values. A comparison with values reported for the study area would be more informative. It is unfortunate that the four landscape classes recognized in this study cannot be linked unambiguously to the surface geology map units, and therefore their proportional extent is unclear. However, given the low representation of the organic class in the study area, it is likely that the mean landscape values for 0-1m and 0-3m are quite similar to those reported in Hugelius et al. (2014).,**

**which are 5-15 KgC m2 for 0-1m (one polygon in the NCSCD) and anything between 20-75 KgC m2 for the 0-3m (two polygons in the NCSCD). They are also quite similar to the Hossain et al. (2015) values reported next.**

Agreed, for the reasons outlined (landscape classes cannot be linked unambiguously to mapped surficial geology) we have chosen to not report mean landscape values. In response to this comment, we have clarified the relationship to mean circumpolar values and now revised (1) the text to clarify: "Mean organic-carbon density in the top 3 m of soil profiles near Lac de Gras is about half of the circumpolar mean values reported in recent statistics for similar soils (Table D1).", and (2) the caption of Table D1 "...for three terrain types from the Lac de Gras study area (Table 2) compared with circumpolar mean values for similar soils reported in a recent compilation (Hugelius et al., 2014, Table 2)."

**P19, L393-394: This is similar to . . . Check sentence**

Has been clarified to "For mineral soils, this is similar to the mean of about 12 (kg C m3) in the northern Canadian Arctic and for organics, it is similar to the mean of about 30 (kg C m3) in the southern Canadian Arctic reported for the top 1 m by Hossain et al. (2015, Fig. 5D)."

**P21, L475-477: Out of scope (see my comment P1, L19)**

[revised manuscript text omitted]